# Micropropagation of Seed-Derived Clonal Lines of the Endangered *Agave marmorata* Roezl and Their Compatibility with Endophytes

**DOI:** 10.3390/biology11101423

**Published:** 2022-09-29

**Authors:** America Martinez-Rodriguez, Celia Beltran-Garcia, Benjamin Valdez-Salas, Fernando Santacruz-Ruvalcaba, Paolo Di Mascio, Miguel J. Beltran-Garcia

**Affiliations:** 1Engineering Institute, Universidad Autónoma de Baja California, Mexicali 21280, Baja California, Mexico; 2Lab 309-E Building, Chemistry Department, Universidad Autónoma de Guadalajara, Zapopan 45129, Jalisco, Mexico; 3Departamento de Producción Agrícola, Centro Universitario de Ciencias Biológicas y Agropecuarias, Universidad de Guadalajara, Zapopan 45110, Jalisco, Mexico; 4Departament of Biochemistry, Institute of Chemistry, University of São Paulo, Sao Paulo 05508-000, Brazil; 5Departamento de Biotecnologicas y Ambientales, Universidad Autónoma de Guadalajara, Zapopan 45129, Jalisco, Mexico

**Keywords:** *Achromobacter*, *Agave marmorata*, clonally propagated, germinated seeds, endophytes, in vitro micropropagation

## Abstract

**Simple Summary:**

The wild *Agave marmorata* Roezl has been classified as an endangered species. Extracting these plants from the forest for commercial purposes and long maturation periods of close to 30 years have contributed to their loss. *A. marmorata* interacts with pollinators and other *Agaves* species to maintain genetic variability. Thus, the conservation and restoration of the agave ecosystem is an ecological challenge. Typically, agave micropropagation use meristem or leaves as explants to rapidly produce uniform agave plants in age and size on a large scale leading to homogeneous plantations. However, introducing these clones to the field reduces genetic variability. This study evaluated in vitro micropropagation of *A. marmorata* from seeds to generate clonal lines. The selected seedlings exhibited variations in multiplication capacity and stable tissue formation. Variations in clonal lines could be exploited to produce high-quality plants with different capacities, such as faster propagation, enhanced stress adaptation, and continued growth under nutrient limitation conditions, consequently maintaining genetic variability. Furthermore, some clonal lines were inoculated with four endophytic bacteria to identify other differences among these plants, including endophyte-host compatibility. Variable responses to inoculation were observed among clonal lines. We found that *Achromobacter xylosoxidans* was compatible, unlike *Enterobacter cloacae* which caused plant death.

**Abstract:**

*A. marmorata* is the raw material used for tepextate mescal production but is classified as an endangered species. In the present study, we obtain and multiply clonal lines of *Agave marmorata* Roezl by selecting seedlings derived from seeds. Ten seedlings from two lots of 400 germinated seeds were selected for axillary bud proliferation induced by BAP 5 mg/L in vitamin-free Murashige and Skoog’s medium. Differences in shoot numbers, heights and senescent tissue formation were observed. Notably, the AM32 line formed 84 shoots and presented low senescent tissue after 60 d of culture. We also selected the AM31 and AM33 clonal lines. Four-month shoots were extracted with 80% methanol in water to determine the total content of saponins, flavonoids, and phenolic acids and compare the three clonal lines. Some bioactive molecules were identified using HPLC techniques and MALDI-TOF mass spectrometry none showed significant differences in content. Additionally, plants derived from the clonal lines were inoculated with four endophytic bacteria. Among these, *Achromobacter xylosoxidans* supported plant growth of AM32. A notable effect of plant death was observed after inoculation with *Enterobacter cloacae,* an endophyte of *A. tequilana*. Additionally, *Pseudomonas aeruginosa*, an endophyte from *A. marmorata*, reduced biomass. Our results demonstrate the incompatibility of *A. marmorata* to *E. cloacae* and specialization between the host plant and its endophytes. The compatibility of the plant-endophyte could be exploited to boost the establishment and stability of mutualisms to benefit plant development, stress tolerance and pathogen resistance. The differences in multiplication capacity, stable tissue formation, and endophyte biotization responses may indicate genetic variability. Clonal selection and micropropagation from seed-derived plants could contribute to conserving the endangered *A. marmorata* plant for reforestation in their natural habitats, thus, assuring mass propagation for sustainable industrial production of mescal, bioactive compounds, and prebiotics.

## 1. Introduction

The landscape of many ecosystems from North to South America and the Canary Islands shows a wide distribution of plants of the genus *Agave* [1]. Agaves are diverse in semi-arid and temperate ecosystems but can grow in humid and mountainous areas. These plants are hardy and capable of growing and reproducing in unfertile soils under low-water conditions for several months. Since ancient times humans have used agave plants as a source of food, drinks, medicines and fibers, and currently, they are a source of prebiotics such as agavins [2,3,4,5]. Notably, more than ten years ago, a great interest in using agave plants and their residues (e.g., leaf fiber) emerged as an alternative to producing sustainable biofuels in Mexico and Australia [6,7,8,9].

On the other hand, most *Agave* species used for mescal production in Mexico are wild and extracted from the forest without planned long-term conservation management [10,11]. During their maturation cycle, agave plants accumulate oligofructan sugars that serve as a reserve material for developing inflorescences, which develop numerous fruits with seeds if those flowers are pollinated. The maturation cycle of agaves can be reached after 5–30 years, depending on the species [12]. However, if pollination does not occur, the development of asexual bulbils is stimulated. It should be pointed out that the collectors commonly extract mature plants before the inflorescences appear. Mature plants produce the highest quality beverage for mescal production, but pollen and seed formation are sacrificed. Consequently, these practices have led many *Agave* species for mescal production to become endangered [13], warranting urgent propagation and conservation strategies.

*Agave marmorata* Roezl, better known as “tepextate or pitzometl”, grows wild in the Tehuacan-Cuicatlan Valley in south-eastern Puebla and north-western Oaxaca states. This species grows in the hills and is part of the vegetation in low deciduous forests. It is a raw material for mescal “tepextate”, a distilled drink of high quality in taste, but their accelerated extraction, long sexual reproduction periods, eroded soils, and climatic alterations have placed this species in a state of “highly threatened”. Without protection from the Mexican Official Norms, *Agave marmorata* Roezl runs the risk of extinction. In addition to mescal production, *A. marmorata* inflorescence and leaves have been used for medicinal and ornamental purposes and as food [14,15,16].

Recently, it was reported that using seeds for plant propagation may increase genetic variability and maximize the repopulation of sexually originated plants such as *Pulsatilla turczaninovii*, *Viola cornuta*, *Echinacea angustifolia* and others [17,18,19,20,21]. However, there is some dispute about whether *Agave* species produce viable seeds for propagation [22,23]. Farmers who use seeds for *A. marmorata* propagation face a problem because seeds are scarce due to plant extraction before inflorescence. Another associated problem is the germination percentage of *A. marmorata* seeds in nature, where seeds have a germination possibility of close to 0.42%, according to Jiménez-Valdés et al. [24], who reported that only three plantlets from 200 seeds were formed. The seed management for propagation purposes must be carefully executed since it has been shown that agave seeds are susceptible to temperature and humidity, which negatively impact germination yields [25,26]. Therefore, in vitro micropropagation of plants derived from seeds might favor economic and environmental purposes for *A. marmorata*.

In vitro micropropagation has become an effective technique for plant production for industrial and medicinal purposes and a strategy to preserve threatened or endangered plants since they can be reintroduced to nature [27,28,29]. Since 30 years ago, an alternative to conventional agave propagation has been using in vitro propagation protocols through plant tissue and cell culture, in which plant growth regulators have been used. The tissue multiplication of *Agave* species has been carried out through different approaches, such as direct organogenesis, forming shoots without the production of previous calluses; indirect organogenesis, where there is the formation of calluses before the shoot; and direct and indirect somatic embryogenesis [30]. The in vitro micropropagation in *Agave* has a wide range of advantages, as recently reviewed by [31]; however, for ecological purposes, the multiplication of identical plants decreases their genetic variability when reintroduced to the forest. On the other hand, not all agave species respond efficiently to plant growth regulators to enhance shoot multiplication, and it also depends on the type of explant used [32,33,34,35]. Delgado-Aceves et al. [36] proposed that the protocols for the multiplication of endangered agave species require modifications of other species to increase the adaptation and propagation effectiveness.

On the other hand, plants derived from in vitro culture to ex vitro conditions must withstand stressful biotic or abiotic conditions. As recently reviewed, plant inoculation with benefic microorganisms positively impacts micropropagation and acclimatization [36]. The inoculation with endophytic microorganisms under in vitro and ex vitro conditions, known as “plant biotization”, can improve stress alleviation, plant growth and support under limited nutrient situations [37,38]. However, the success of biotization relies on plant-microbe compatibility.

This work aimed to obtain and efficiently propagate clonal lines of *A. marmorata* from seeds based on the natural capacities of seed germination to plant establishment (individualization) without physical or chemical induction. The total content of flavonoids, phenolics, and saponins was measured in shoots of the three selected lines, and some bioactive compounds were separated and subsequently identified by HPLC-MALDI-TOF. Additionally, we evaluated the inoculation of clonal lines with four agave seed-vectored endophytic bacteria to determine plant-microbe compatibility and their effect on ex vitro plant biotization to facilitate the future reintroduction of *A. marmorata* plants for conservation and restoration purposes.

## 2. Materials and Methods

### 2.1. Seed Material

Seeds were harvested from mature fruit capsules of wild *Agave marmorata* plants before their dehiscence in the municipality of Tlacolula de Matamoros in the central Oaxaca Valley, Mexico, with the geographical coordinates 16°57 13′ N, 96°28 33″ W by the biologist Francisco Echaide-Aquino in April 2015. Black seeds were transported to the laboratory in paper bags and stored in the dark at 25 °C, and white seeds were discarded. All germination assays were carried out in Laboratory 309 of the Biotechnology and Environmental Sciences Department at the Autonomous University of Guadalajara, in Guadalajara Jalisco, Mexico, from May 2015 to November 2016.

### 2.2. Surface Disinfection of Seeds

Seeds were soaked in 1% (*v*/*v*) Extran M02 detergent solution (Merck, Tizapan, San Angel, Mexico) for 10 min to disinfect the surface and remove organic residues, dust, and epiphytic bacteria. Then, the seeds were disinfected in a 3% sodium hypochlorite solution (commercial Clorox, 6% *w*/*w*) with constant agitation in a laminar flow hood (Esco, Changi, Singapore), using metal tea infusers immersed for 10 min. The seeds were then submerged in 80% ethanol for 5 min and washed three times with sterile distilled water for 10 min to remove any chlorine or ethanol residues. Finally, 100 µL of the wash water were plated directly onto a trypticasein Soy (TSA) agar (BD Bioxon, Beckton Dickinson, Ciudad de México, Mexico) to determine the presence of bacteria.

### 2.3. Seed Germination Experiments

Groups of cleaned seeds (n = 12) were transferred into a Petri dish amended with 2% Phytagel medium (Sigma-Aldrich, St. Louis, MO, USA) and incubated at 30 °C in an artificial climate incubator (Thermo Fisher Scientific, Waltham, MA, USA) with a 12 h photoperiod under LED illumination (22 µmol m^−2^ s^−1^). For each evaluation, seed germination (200 seeds per lot) was analyzed in triplicate in May and November of 2016. Two lots of seeds were utilized, one collected and stored for 12 months and the other for 18 months, referred to herein as L12 and L18. Replicates of each trial were made no more than three days between each run, and no seedlings were selected from replicates. The germination data were recorded every 24 h. The criteria for seedling selection included radicle length 1 mm out of the seed coat and the cotyledon visible at maximum on the fourth day after imbibition. The germination stopped when no new seed germination was observed for 15 consecutive days.

### 2.4. Shoot Multiplication

Ten seedlings derived from selected germinated seeds were multiplied using the axillary shoot method. Briefly, the roots and the distal part of the leaves were removed, and one single seedling per jar was cultured with 25 mL with 15 g/L of bacteriological agar of solid modified Murashige and Skoog [39] medium without vitamins plus 5 mg/L [40] of the plant growth regulator 6-benzyl aminopurine (BAP, Sigma-Aldrich). Seedlings were incubated in a culture room at 30 °C with 12 h a photoperiod under LED illumination (22 µmol m^−2^ s^−1^).

The MS medium consisted of a macronutrient stock containing: 165 g/L^−1^ (Na_4_)NO_3_, 37 g/L ${\mathrm{MgSO}}_{4}\cdot {7H}_{2}0$, 190 g/L ${\mathrm{KNO}}_{3}$, 17 g/L ${\mathrm{KH}}_{2}{\mathrm{PO}}_{4}$, a micronutrient stock containing: 1.240 g/L $H_{3}{\mathrm{BO}}_{3}$, 3.38 g/L ${\mathrm{MnSO}}_{4}\cdot {4H}_{2}O$, 2.12 g/L ${\mathrm{ZnSO}}_{4}\cdot {7H}_{2}O$, 5 mg/L ${\mathrm{Na}}_{2}{\mathrm{MoO}}_{4}\cdot {2H}_{2}O$, 5 mg/L ${\mathrm{CuSO}}_{4}\cdot {5H}_{2}O$, 5 mg/L ${\mathrm{CoCl}}_{2}\cdot {6H}_{2}O$, 166 mg/L KI, a calcium chloride stock: 20 g/L ${\mathrm{CaCl}}_{2}\cdot {2H}_{2}O$ and an Fe∙EDTA stock: 2.784 g/L ${\mathrm{FeSO}}_{4}\cdot {7H}_{2}O$, 3.724 g/L ${\mathrm{Na}}_{2}\cdot \mathrm{EDTA}$. All reagents were purchased from Sigma-Aldrich.

The final base medium was composed of 10 mL of the macronutrient stock, 5 mL of the micronutrient stock, 22 mL of the calcium chloride stock, 10 mL of the Fe∙EDTA stock, 30 g/L sucrose, 7.5 g/L agar and water to complete one liter. All materials and culture media were autoclaved at 121 °C for 20 min.

Once the multiplied shoots were obtained, morphological evaluations of tissue state (tissue with oxidation, formation of crystalline shoots, presence of roots and senescence of leaves), coloration (green, purple, albino) and the number of shoots formed were utilized for clonal line selection. After two months of culture, the number of shoots induced per plant and the length of shoots were determined. The shoot multiplication was evaluated five times. Each baby food jar had an initial number of 5 shoots per experimental replicate, and the new shoots were counted. The final growth data and the number of generated shoots were recorded after two months. Once a clonal line was multiplied and selected, it was placed in an acclimatization medium free of plant growth regulators for 30 d for root growth.

### 2.5. Plant Adaptation to Beach Sand-Vermiculite Microcosm Nitrogen-Free

Shoots were separated and placed in magenta boxes for rooting in 120 mL of MS medium containing 0.025 mg/L kinetin, 30 g/L sucrose, 2 g/L activated charcoal and 15 g/L bacteriological agar as the gelling agent. Shoots were cultured on rooting media for 100 d under a 12 h photoperiod with LED illumination (22 µmol m^−2^ s^−1^). Well-developed plantlets per clonal line with a size of 3–4 cm and approximately 2 g were acclimatized in a glass flask containing a mixture of autoclaved vermiculite and beach sand (5:50) irrigated with saline MS medium without nitrogen and sugar sources. It should be mentioned that the beach sand collected at “Punta Perula” beach in la Huerta Jalisco, Mexico (Latitude: 19.587855, Longitude: 105.128493) was extensively washed with tap water to remove organic material, air-dried, and autoclaved for 3 h. These plants were grown in a culture room at 30 ± 2 °C. The survival rate of the plants was determined after eight weeks.

### 2.6. Phytochemical Compounds Analysis of Selected A. marmorata Clonal Lines

#### 2.6.1. Plant Material and Its Lyophilization

Six-month-old in vitro-derived shoots from clonal lines AM31, AM32, and AM33 were used as the plant material for phytochemical extraction. Residues of agar, dry plant tissue and roots were carefully removed. The plant material was weighed and adjusted to 10 g per sample and immediately frozen with liquid nitrogen for 30 min. Shoots were pulverized in a cold mortar, placed on ice, and then freeze-dried. Lyophilization was performed in a LabCONCO Free Zone 2.5 plus (LabConco Corp, Kansas, MO, USA) with a vacuum condition of 0.10 mbar at −85 °C for 11 h. Additionally, 5 g of *A. marmorata* leaves stored in pots and soil under greenhouse conditions for three years were lyophilized in the same way as the shoots to obtain a fine dry powder for phytochemical extraction.

#### 2.6.2. Sample Preparation to Obtain Extracts

For phytochemical evaluation, lyophilized shoots and agave leaf powder (400 mg) were ground for 3 min in a mortar and pestle and sequentially extracted with 15 mL of the following solvents: (a) 0.5% trifluoroacetic acid in methanol, (b) ethanol:water (50:50 *v*/*v*) and (c) methanol:water (80:20 *v*/*v*). Extracts were shaken in a rotary incubator (Infors HT, Ecotron, Rittergasse, Bottmingen, Switzerland) at 150 rpm for 48 h at 25 °C, and plant residue was removed by centrifugation at 6300× *g* for 10 min at 4 °C. Residues from each solvent were collected in the same vial and stored at −80 °C. The supernatant was collected and evaporated in a vacuum. For the HPLC-PDA and UHPLC-UV analyses, 50 mg of the shoot extracts were reconstituted in hydroalcoholic extract comprised of water:ethanol:methanol (10:30:60 *v*/*v*). All reagents were purchased from Sigma-Aldrich.

#### 2.6.3. Total Flavonoid, Phenolic and Steroidal Saponin Quantification

Total flavonoid content in *A. marmorata* shoot extracts was determined using the aluminum trichloride (AlCl_3_) method described by Chang et al. [41]. Initially, 200 µL of each sample (10.0 ± 0.1 mg dissolved in 1.0 mL of 80% methanol), 400 µL of water and 1000 µL of AlCl_3_ solution (prepared as follows: 133 mg crystalline aluminum chloride, 400 mg crystalline sodium acetate dissolved in 100 mL of 80% methanol) reagent was added. Absorbance was recorded at 430 nm using an Ultraviolet-Lambda 25 spectrophotometer (Perkin Elmer, Waltham, MA, USA). Total flavonoid content was calculated based on a calibration curve (y = 4.9218x − 0.0024, r^2^ = 0.9997) using quercetin (5–50 µg/mL) solutions in 80% methanol (Sigma-Aldrich). The total flavonoid content is expressed as mg of quercetin equivalent per gram of extract dry weight (mg QE/g DW).

Total phenolics were determined using the Folin–Ciocalteu reagent with gallic acid as the standard (Sigma-Aldrich). Samples (500 µL) were mixed with 250 µL of 50% Folin-Ciocalteau reagent and allowed to stand at room temperature for 5 min. Then, 2000 µL of a sodium bicarbonate solution (7% *w*/*v*) were added to the mixture. After 90 min at room temperature, absorbance was measured at 760 nm using a UV/Vis spectrophotometer. Total phenolics were quantified using a calibration curve (y = 54.554x − 0.0016, r^2^ = 0.9959) obtained from a known concentration of the gallic acid standard (2–200 µg/mL). The concentration is expressed as milligrams of gallic acid equivalents per gram of dry weight (mg GAE/g DW).

The steroidal saponin content was calculated using a diosgenin standard curve (y = 3.503x + 0.0179, r^2^ = 0.9965) obtained from known concentrations (5–150 µg/mL) of diosgenin (Sigma-Aldrich). The sample solution was prepared in water (at 10.0 ± 0.1 mg/mL), dissolved in an equal volume of ethyl acetate (Merck), and agitated twice for 1 min on a vortex. Next, 500 μL of crude extract and 500 μL of 0.5% anisaldehyde (Sigma-Aldrich) reagent were mixed and kept for 10 min. Then, 2 mL of 50% sulphuric acid (Golden Bell, Mexico) in ethyl acetate reagent was added and mixed. Samples were placed in a 60 °C water bath for 10 min, and then the absorbance was recorded at 430 nm. The concentration of total steroidal saponin is expressed as mg diosgenin equivalents per gram of dry weight extract (mg Diosgenin E/g DW).

The total saponin content was calculated using a diosgenin standard curve (y = 0.6509x + 0.283, r^2^ = 0.9953). Ten mg of crude extract were dissolved in 5 mL of 50% aqueous methanol. Then, 250 μL of the aliquot was transferred to test tubes into which an equal volume of vanillin reagent (8%) was added, followed by 2.5 mL of 72% (*v*/*v*) sulphuric acid. The mixture was mixed and placed in a 60 °C water bath for 10 min. The tubes were cooled in an ice-cold water bath for 3–4 min, and the absorbance of the yellow color reaction mixture was measured at 544 nm against a blank containing 50% aqueous methanol instead of a sample. The total saponin concentration is expressed as mg Diosgenin equivalents per gram of dry weight extract (mg Diosgenin E/g DW).

#### 2.6.4. HPLC-PDA Analysis/Fraction Collection UHPLC-UV

The phytochemical compounds extracted from the shoot and leaf tissue were analyzed using an HPLC coupled to a photodiode detector (Waters e2695 HPLC system with a 2998 PDA detector, Milford, MA, USA) and a Luna C18 (2)-100A, 4.6 × 250 mm (5 µm) column (Phenomenex, Torrance, CA, USA). The gradient elution consisted of mobile phase A (0.5% trifluoroacetic acid in water) and mobile phase B (0.5% trifluoracetic in acetonitrile) at a flow rate of 1.0 mL/min. The gradient was run as follows: mobile phase B was maintained at 5% for 2.5 min, mobile phase B increased from 5% to 23%; from 2.5 to 35 min, mobile phase B (isocratic) was fixed at 23% from 35 to 40 min, mobile phase B increased from 23% to 85% from 40 to 78 min, mobile phase B (isocratic) fixed at 85% from 78 to 83 min, mobile phase B fixed at 90% from 83 to 90 min, and finally, mobile phase B returned to 5% and was maintained from 90 to 95 min. The sample volume was 50 µL at an equal concentration of 30 mg/mL (±0.01). Absorbance peaks at λ = 518 nm (for anthocyanins), λ = 205 nm (for saponins), λ = 275 nm and λ = 390 nm (flavonoids and phenolic acids) were measured and recorded.

An HPLC with variable length UV-Vis detector (Ultimate 3000, Thermo Scientific) coupled to an auto-fraction collector was used to collect separated fractions under the same chromatographic conditions described for the HPLC-PDA analysis. The sample volume was 50 µL at an equal concentration of 32 mg/mL (±0.01). Fractions were collected simultaneously at the four wavelengths selected (λ = 518 nm, λ = 205 nm, λ = 275 nm and λ = 390 nm and then lyophilized for MALDI-TOF mass spectrometry analysis.

#### 2.6.5. MALDI-TOF Mass Spectrometry Analysis

Dried fractions were resuspended in 100 μL of HPLC grade methanol and homogenized on a Vortex apparatus before analysis. MALDI-TOF mass spectra were obtained using 2,5 dihydroxybenzoic acid (DHB) as the matrix (2,5 DHB 30 mg, 990 μL acetonitrile, 10 μL trifluoroacetic acid) on an Autoflex speed MALDI-TOF-TOF (Bruker Daltonics GmbH, Bremen, Germany) equipped with a 355 nm pulsed nitrogen laser. Peak acquisition was in the positive and negative reflectron mode, using an accelerating voltage of 20 kV. The mass range for analysis was 350–1500 Da.

One μL of each resuspended sample was spotted onto the steel MALDI plate and dried at room temperature for 5 min. Then, 1 μL of the matrix DHB was added to each well and dried again. Mass calibration was achieved with peptide standards (Peptide Calibration standard II, Bruker Daltonics) spotted onto a stainless-steel target plate before data acquisition. The Flex Analysis 3.3 software (Bruker Daltonics, GmbH, Bremen, Germany) processed mass profiles to produce each fraction’s peaks (*m*/*z*). Default settings were used for the baseline correction and smoothing.

### 2.7. Plant Growth Stimulation with Endophytic Bacteria

Ten plants for each treatment of the three clonal lines (AM31, AM32 and AM33) were used to evaluate the effect on plant growth stimulation and compatibility of seed-transmitted endophytes of *A. marmorata* and *A. tequilana* such as *Achromobacter xylosoxidans* (*Ax*) N-strain (GenBank access MF540449.1), *Pseudomonas aeruginosa* (*Pa*) EP60A3 (GenBank access MF540452.1); *Bacillus tequilensis* (*Bt*) AT-10 (GenBank access KY681426.1) and *Enterobacter cloacae* SEA01 (GenBank access KY625189.1).

The Table 1 summarizes the plant growth promotion characteristics of the endophyte bacteria used to evaluate their growth stimulation on clonal lines. ACC (ACC deaminase, [42]), IAA (Indole Acetic Acid, [43]), N-BRIP (phosphate solubilization, [44]), NFB (Nitrogen Fixing Bacteria, [45]).

For inoculum production, the bacterial strains were cultured in TS broth (Trypticasein Soy) and incubated at 32 °C with agitation of 100 rpm for 20 h. Bacterial suspensions were centrifuged for 5 min at 12,850× *g*. The supernatants were discarded, and the cell pellets were washed three times with 0.9% saline solution plus 0.1% glucose. The bacterial pellet was then suspended in the saline-glucose solution at an OD_600_ of 1.0 equivalents corresponding to *Bt* 70 × 10^6^, *Ec* 17.6 × 10^6^, *Ax* 1.5 × 10^6^ and *Pa* 3.4 × 10^6^ CFU/mL.

Each acclimatized plant received 2.5 mL of bacterial inoculum directly in the substrate near the roots. Five inoculations were made to each group of plants per clonal line over a 14 d period (i.e., 23, 25 and 27 May 2018 and final inoculations on 1 and 5 June 2018). Plants were grouped as follows: (G1) water control, (G2) *B. tequilensis* (Bt), (G3) *E. cloacae* (Ec) isolated from *A. tequilana*, (G4) *A. xylosoxidans* (Ax) and (G5) *P. aeruginosa* (Pa) isolated from *A. marmorata*. Plants were maintained under the same conditions described previously for acclimatization. After the final bacterial inoculation, plants were irrigated with 2.5 mL sterile distilled water every 3 d. Thirty days post-inoculation (from the first inoculation), the number of new leaves, roots, root length and fresh and dry weight of the plants were recorded.

### 2.8. Statical Analysis

Data analysis and statistics were performed using Excel 2013 (Microsoft, Albuquerque, NM, USA). All results are expressed as the mean ± standard deviation (SD), n = 3 for all experiments. The Statgraphics V. Centurion XVII.2 software (Statpoint Technologies Inc., Plains Virginia, VA, USA) was used to conduct multivariate ANOVA with multiple range tests for the shoot and plant growth data. Multiple comparisons were performed using the Tukey methods. The level of significance was set at *p* ≤ 0.05.

## 3. Results

### 3.1. Seed Germination Rate and Final Germination Percentage

In this study, we established a micropropagation protocol for *A. marmorata* using selected seedlings derived from those germinated seeds with visible radicles in a maximum of 4 d post-imbibition, which will be used to obtain different clonal lines for subsequent multiplication.

Two lots of 200 seeds after 12 and 18 months of storage (L12 and L18) were germinated in triplicate with 3 d between replicates to obtain seedlings. The germination distribution is shown in Figure 1A. The L18 seeds had more germination in the first three days than those from L12 (22 seeds vs. 1 seed). However, the percentage of maximum accumulative germination distribution was greater for L12, increasing from 44% to 87% between days 4 to 8. The number of germinated seeds from L18 also increased slightly on day 6 by 52%, which is equivalent to 104 germinated seeds. This result was the highest one-day yield for this lot of seeds. There was a decrease in the daily germination rate on day 9 in both seed lots. The total accumulated germination of L18 was about 77% on day 10, and L12 had 90.25% on day 9.

On the other hand, Figure 1B represents the germination rate until day 9 for L12 and day 10 for L18. Seedlings derived from seeds germinating on or before day 4 were selected as the most optimal material for in vitro micropropagation.

For the micropropagation assays, six L18 and four L12 seedlings were utilized (Table 2). Although many seeds completed their germination within 4–6 days, few seedlings reached an appropriate state for selection due to the size and number of roots and/or limited cotyledonary leaves. The selected seedlings evaluated herein formed cotyledon leaves between days 4 and 6 and two true leaves 60 d post-germination. Approximately 50% of germinated seeds produced seedlings that stopped developing within a few days.

### 3.2. Seedling Selection for Clonal Propagation

Once the germination assay was concluded, 70 seedlings were analyzed for micropropagation by culturing seedlings derived from seeds in MS+BAP (at 22.2 µM) medium. The following characteristics: appearance/shoot morphology, tissue coloration, spontaneous rooting, malformations, and susceptibility to oxidation were used for clonal line selection.

As examples of shoot appearance, Figure 2A shows shoots of the AM32 line, which displayed rapid tissue adaptation and well-defined shoots and leaves. Figure 2B shows the AM31 line with hyperhydric or crystalline buds (previously known as vitrification). This tissue is prone to rooting and forming shoot clusters that are easily detached and individualized. The AM31 line also had curly leaves with glassy, fragile, deformed shoots containing excess water (Figure 2C). Typically, excessive water accumulation alters in vitro propagation; however, tissue from the AM31 line adapted later.

Concerning tissue coloration, three main color tones were observed: totally bright green (Figure 2D), purple tones (Figure 2E) and white (chlorosis) (Figure 2F). Purplish tones were evident in the AM31, AM33 and M14D-E lines. In their natural habitat, young *A. marmorata* plants have a purplish coloration; thus, the shoot coloration provides information about plant identity. Chlorotic shoots and leaves were observed in nine lines numbered AM50 to AM58 (Figure 2F).

The physiological state of the shoot tissue was evaluated by monitoring phenolic oxidation or browning of tissues, root production and senescence, which influence nutrient uptake and/or can contribute to premature death. As shown in Figure 2G, AM32 and AM34 showed a browning thick layer of crust tissue that had to be manually removed for further budding and new shoot tissue formation. Several lines formed roots, for example, the AM20 and AM31 lines exhibiting abundant root production (Figure 2H). The formation of roots is a crucial factor for adaptation to an ex vitro system, but under in vitro conditions, it cancels the effect of shoot induction.

A representative image of senescent leaf tissues is presented in Figure 2I. The M06, AM36 and AM38 lines displayed these features and could not form vigorous shoots. These lines also had a high tendency for individualization, attenuated shoot formation and low survival. Thus, utilizing these lines for mass plant production would be impossible. The lines that presented any of these negative characteristics were discarded. Sixty lines were ultimately eliminated because they could not correctly adapt for in vitro propagation.

Only ten clonal lines derived from germinated seeds were selected for micropropagation. Table 1 summarizes the growth and development parameters of the ten selected seedlings. As the main characteristic, the selected seedlings exhibited large healthy cotyledon leaves (length and width) and enhanced root development. Furthermore, the selected seedlings were vigorous, turgid, and bright green.

### 3.3. Shoot Multiplication Induction with BAP in MS without Vitamins

Next, we used the axillary bud proliferation method to compare the shoot multiplication of ten selected seedlings (lines). This method involves applying one concentration of BAP cytokinin (5 mg/L or 22.2 µM) as a shoot inducer in a vitamin-free MS medium for eight weeks to understand the natural tissue response for proliferation between lines stimulated by BAP, rather than determine what would be the best concentration of BAP for shoots induction. Seedlings in MS medium without BAP did not multiply shoots; however, the propagule developed a “mature plant” that did not grow (Appendix A). The shoot multiplication assay was repeated five times for each of the ten clonal lines. In addition to the number of shoots formed, the average shoot length and senescent tissue (dead). As shown in Figure 3, the formed shoots were green and purple according to the bud ripening.

Furthermore, in Figure 3A, the AM32 line had the highest average number of adventitious shoots, with 83.4 ± 4.56. The clonal lines AM21, AM23, AM32 and AM33 were statistically different in producing shoots among the lines and formed the first group with more than 30 shoots. The clonal lines AM33, AM23 and AM31 formed 43 ± 12.23, 39.6 ± 2.07 and 32 ± 7.45 shoots. The M14d-e and AM20 lines formed a second group, with 24.6 (±2.70) and 25.8 (±6.50) adventitious shoots without significant differences. Lastly, the third group with less than 20 shoots formed includes four lines: AM34, M06, AM21 and AM24 (14.8 ± 2.77, 13 ± 2.12, 12.5 ± 10 and 6.6 ± 2.07). This group was not statistically different (Tukey *p* ≤ 0.05). The AM24 displayed low shoot production.

Concerning shoot length results, we observed that clonal lines also could be divided into three groups. The M14d-e line presented the longest shoots at 1.94 ± 0.29 cm, followed by AM20 (1.76 ± 0.34 cm) and AM33 (1.86 ± 0.47 cm), without significant differences between them. In the second group AM31 had 1.36 ± 0.59, AM32 had 1.1 ± 0.22, and AM34 had 1.04 ± 0.09. Some lines showed a length of shoots less than 1 cm, including the M06, AM21, AM23, and AM24, with values of 0.9 ± 0.42, 0.5 ± 0.07, 0.6 ± 0.12 and 0.36 ± 0.13 cm, respectively (Tukey *p* ≤ 0.05). The line with the highest average number of shoots, AM32, produced short shoots. On the other hand, the AM33 and AM31 lines produced 50% and 38% fewer shoots than AM32, but the shoots were 70% and 23.63% longer.

Another criterion for choosing clonal lines for micropropagation was senescent tissue formed based on the total shoot number formed. With this data, we can visualize the rate of senescence experienced by the shoot tissue in each line for eight weeks (Figure 3C). The AM24 and M14d-e lines exhibited the most and least amount of senescent tissue, respectively. The senescent tissue was not different between the AM32 and AM33 lines, and the AM31 line had significantly more.

In addition, the M06, M14D-E, AM20, AM24 and AM34 clonal lines formed brown shoots, which limited their proliferation since the growth points were not stimulated due to a large amount of dead tissue. Spontaneous rooting was observed in the AM21, AM24, and AM34 lines, indicating that these plants tend to individualize easily.

After the shoot proliferation experiment, the main parameter to select the lines AM31, AM32 and AM33 for biotization assays was the formed shoot number and the significative difference among the other lines. Elongated shoots were transferred to magenta boxes with 120 mL of MS medium plus 0.025 mg/L of kinetin, 30 g/L sucrose, plus 2 g/L of activated charcoal for rooting. Explants were incubated for 100 d under 12 h photoperiod under LED illumination (22 µmol $m^{-2}\;s^{-1}$). After rooting, six plantlets of the AM31, AM32 and AM33 were transferred to glass flasks containing a mixture of autoclaved vermiculite and beach sand (5:50) and grown in a culture room at 25 ± 2 °C. The survival rate of the plants was 100%.

### 3.4. Comparative Analysis of Total Flavonoid, Phenolic and Saponin Content in the Clonal Lines

We determined the total phytochemical content in the formed shoots to identify differences among the selected seed-derived clonal lines. For comparison, the phytochemical content in the leaf of three-year *A. marmorata* “Hijuelo” or HAM, separated from the mother plant and maintained in pots with oak soil, was also assessed. The results show slight differences between the phytochemical and the lines (Table 3). For example, in the total phenolic content, only AM31 was different from the other two lines and different from HAM. No significant differences were observed between lines for the total flavonoids, and only HAM was different. We observed that the total flavonoid content of the potted plant was 8.35 times less than AM31. There were no significant differences in all tissues analyzed when the total content of both saponins was determined.

For HAM, the phenolic compound and saponin concentrations fell into the same range reported in micropropagated shoots (Table 2).

### 3.5. MALDI-TOF Profiling of the HPLC Collected Fractions from Clonally Propagated Lines of A. marmorata

HPLC-PDA and HPLC-UV methods were developed to detect differences in the concentrations and types of secondary metabolites accumulated in the shoots of the three *A. marmorata* lines. The HPLC separation of extracts from each clonal line was monitored at four wavelengths: 205 nm, 275 nm, 390 nm, and 518 nm. According to the wavelengths monitored and total phytochemical content analysis, it was determined that approximately 66 fractions were separated. Differences in chromatographic profiles of hydroalcoholic extracts were expected; however, each line’s number of separated fractions was similar, only presenting variations in the intensities and peak areas.

Seventeen fractions were separated at 275 nm (F2, F5, F7, F8, F12, F15, F22, F28, F29–F32, F37, F42, F44, F47, F48); three fractions at 205 nm (F1, F14, F62), two fractions at 390 nm (F25 and F66) and three fractions at 518 nm (F18, F23, F24). Table 4 presents those fractions/compounds that differed between lines according to area percentage normalization and summarizes the chromatographic characteristics and *m*/*z* ion assignments of anthocyanin, flavonoids, tannin, and phenolic acids.

The fractions were analyzed for tentative identification by MALDI-TOF mass spectrometry. Phenolic compounds present in the fractions that absorbed at 275 nm included: Catechin (F2, [M+] at *m*/*z* 290 and [4M+H+K] at *m*/*z* 1201), Gallic Acid (F5, [M+] at *m*/*z* 171), Caffeic Acid (F15, [M−H] at *m*/*z* 180) and Sinapic Acid (F28, [M+] at *m*/*z* 224), while the flavonoids were represented by Isorhamnethin Rutinoside (F7, [M+H+Rutinoside] at *m*/*z* 635), Isorhamnetin-Glucuronide (F42, [M+H+Glucuronide] at *m*/*z* 496), Isorhamnetin 3-Glucoside (F44, [M+H+Glucoside] at *m*/*z* 473), Isorhamnetin-3-O-Diglucoside (F32, [M+H+2 Glucosides] at *m*/*z* 635), Myricetin Glucoside (F8, [M+H] at *m*/*z* 482), Myricetin 3-O-Glucuronide (F12, [M+H] at *m*/*z* 496), Myricetin 3-O-Rutinoside (F14, [M+H+Rhamnosy-Glucoside] at *m*/*z* 628), Rutin (F22, [M+H] at *m*/*z* 607), Quercetin-3-O-Diglucoside (F29, [M+H+2 Glucoside] at *m*/*z* 621), Quercetin-3-Glucoside or Isoquercitrin (F30, [M+H+Glucoside] at *m*/*z* 465) and Kaempferol Diglucoside (F31, [M+H+2 Glucoside+Na] at *m*/*z* 628).

Based on the percentage of the normalized area (NA) (Table 4), AM31 (λ = 275 nm) had lower levels of Isorhamnethin Rutinoside (F7), Myricetin 3-O-Glucuronide (F12), Caffeic acid (F15), Isorhamnetin-Glucuronide (F42), Isorhamnetin 3-Glucoside (F44) and the probable saponins (F47–F48) than the other two lines. In contrast, AM31 had more Myricetin-3-Glucoside and Quercetin-3-O-Diglucoside than AM32 and AM33. Indeed, these results represent the most remarkable differences among the three lines. We could not identify F1 collected at 205 nm, but its contents were higher in AM32 (NA 24.19%). Thin-layer chromatography (TLC) analysis indicates that this fraction is a mixture of compounds. Moreover, F14 and F62 (Myricetin 3-O-Rutinoside and Myricetin 3-O-Glucuronide, respectively) were increased in AM31.

The fractions F25 and F66 (λ = 390 nm) tentatively corresponded to the flavonoids Quercetin-Glucuronide (F25, [M+H+Glucuronide+Na] at *m*/*z* 484) and Kaempferol Rutinoside (F66, [M+H] at *m*/*z* 595). At 518 nm, fractions contained anthocyanins and hexoses designated as glucose. The AM31 line had the highest Cyanidin 3,5 Diglucoside (F23, NA 35.48%, [M+H] *m*/*z* at 613) content compared to AM32 (NA% 23.08) and AM33 (NA%21.66). Delphinidin 3,5-O-Diglucoside (F24, [M+H] *m*/*z* 656) was slightly higher in the AM31 line. The F18 corresponded to Delphinidin-3-O-Glucoside ([M+Glucoside+Na] at *m*/*z* 484) and was slightly elevated in AM32. The HPLC results were generally consistent with the total phytochemical content analyses since the concentrations adjusted to area normalization were similar among clonal lines. Interestingly, at least 30 fractions were absent from the *A. marmorata* hijuelo leaf extract compared to micropropagated shoots, many being anthocyanins and flavonoid-type compounds.

In our study, ionization was better in reflectron positive mode; however, many compounds analyzed here also ionized efficiently in negative mode. The analyzed samples were co-crystallized with DHB to increase the efficiency and reproducibility of ionization. The certainty of identifying the molecules obtained from the shoot extracts was increased using commercial standards. All the molecules identified in this work have been reported in other agave species, with some being associated with biological properties such as disease control and new drug development for chronic and acute diseases [3].

The MALDI-TOF spectra of extracts from the three clonal lines standardized to the same concentration (10 mg/mL) are presented in Figure 4. The extracts had similar ion fragment (*m*/*z*) profiles with slight intensity variations and signal appearances at *m*/*z* 1361.978 in the AM33 line (Figure 4C). This molecule could not be identified, but its *m*/*z* ion fragment suggests it is a saponin-like molecule.

### 3.6. Plant Growth Parameters and Compatibility of Three Clonal Lines with Endophytic Bacteria

Our intention in applying endophytes was to assess their compatibility and effects on plant growth parameters between clonal lines. We evaluated the effect of an intensive inoculation with four endophytic bacteria on plants of the three selected *A. marmorata* clonal lines. Figure 5 shows the growth results, including the emergence of new leaves and number of formed roots, root length (cm), and fresh and dry weight (g) of plants inoculated with endophytic bacteria. During this assay, plants were not nourished with any mineral solution or carbon or nitrogen source. As expected, no significant effect was observed on plant growth in any clonal line under control treatment irrigated saline-glucose solution. Therefore, the positive and negative effects on plant growth must be attributed directly to bacterial inoculation.

Figure 5 shows that the AM31 line does not respond significantly to bacterial inoculation. For example, no new leaves emerged in plants treated with *B. tequilensis* and *P. aeruginosa*. Similarly, no response in new root formation was detected following microbial treatments, except for plants inoculated with *A. xylosoxidans* (*Ax*), which formed three more roots than the control. In the AM31 line, *E. cloacae* inoculation caused plant death. Inoculating AM32 with *B. tequilensis* (*Bt*) or *A. xylosoxidans* (*Ax*) induced the emergence of one or two new leaves during the post-inoculation period. Control plants did not form leaves. The leaves of plants inoculated with *Ax* were dark green. The highlighted effect of bacterial inoculation on AM32 is related to the number of roots and their length. For example, *Ax* increases up to 14 new roots compared to control plants. For the *Bt* strain, a total of 11 new roots was observed, followed by *Pa* with 8 and *Ec* with 3. An effect on root length was associated with the *Ax* and *Pa* strains, yielding nine and seven centimeter increases.

Although the AM32 line had the most outstanding response to bacterial inoculation, this is not reflected in the total biomass acquisition. The highest dry weight was observed for *Ax* treatment with 3.82 g. Negative data were observed for *Ec*, *Pa*, and *Bt* with −3.08 g, −2.98 g, and −1.95 g, respectively. For AM31, no significant differences were observed between *Ax* and *Bt* (1.32 g and 1.20 g) inoculation in this parameter; however, a negative biomass increment was observed for bacterial treatments with *Pa* and *Ec* with −1.40 g and −0.78 g.

Like the other two clonal lines, endophytic inoculation increased root length and number of AM33 compared to the control. *Bt* induced the formation of 10 new roots, with an average close to 11 cm. In the treatments with *Ax* and *Pa*, we observed the formation of 5 and 8 roots more than the control, respectively. No significant changes in weight gain were detected. The *A. xylosoxidans* strain contributed the greatest to root formation in all clonally propagated lines, producing 16 new roots. Here, we found a notable effect on the plant death caused by *E. cloacae* inoculation and a reduction in biomass acquisition of the AM31 and AM32 lines caused by *P. aeruginosa*. These observations indicate incompatibility between *E. cloacae* and the clonal lines and specificity of *A. xylosoxidans* and *P. aeruginosa* for *A. marmorata*.

*Bacillus tequilensis* is an endophytic bacterium contained in the seeds of *A. tequilana*. It has a good effect on the AM33 line, especially concerning root formation (number and length). In the other clonal lines, its effects were quite limited. Herein, the AM32 line had a moderate response to the inoculation of this bacterium, and some plants died.

## 4. Discussion

### 4.1. Seedling Selection Derived from A. marmorata Seed Germination Capabilities

In this paper, we analyzed the germination capacities of more than 400 seeds and the seedlings’ development as parameters that allowed us to select clonal lines of *A. marmorata*. Subsequently, the three lines selected for their propagation capabilities were analyzed in terms of their total phytochemical content, with some flavonoids and phenolic compounds identified by mass spectrometry. However, the main difference was the compatibility response presented by the lines to the inoculation of endophytic bacteria native to *Agave* species, including *A. marmorata* and *A. tequilana.*

Here, as starting material for micropropagation, we selected seedlings derived from seeds with over-protruding capacities both in germination and in forming a stable tissue to have clonal lines of *A. marmorata*, an endangered species. The first feature for our selection of clonal lines was to avoid masking the natural seed germination vigor using chemical or mechanical treatments of scarification or nutrient use in the phytagel agar that could influence germination. According to germination results, both lots of seeds kept in the laboratory for 12 and 18 months presented germination percentages of around 90% in less than ten days (Figure 1). However, the selected plant lines come from seeds that had completed germination in 4 days (Table 1). This parameter was used to make a more selective selection of the seedlings. The *A. marmorata* germination percentage was higher than the 30% reported for *A. tequilana* seeds [23]. In 2021, a seed collection carried out by our group in *A. tequilana* fields from Jalisco, Mexico, found that their germination percentage was not greater than 20% after two months. Concerning other Agave species, the seeds of *A. marmorata* used in this work presented superior germination capacities. For example, a study with *A. salmiana* reported 93% germination after day 11 of imbibition in seeds subjected to mechanical and chemical scarification [46]. In other work, seeds of *A. grijalvensis* had germination percentages between 15% and 24% after treatment with 20% H_2_O_2_, 0.5% agrimycin-captan, 70% ethanol, 0.1% HgCl_2_, 3% calcium hypochlorite as disinfectant agents [47]. Additionally, it is important to note that the germination percentages of the *A. marmorata* seeds, germinated after 12 and 18 months after collection and storage in the laboratory at 25 °C, were reduced by 8% and 21% compared with the assays conducted in May 2015, which had 98% germination.

### 4.2. Seedling Response to MS+BAP for Clonal Selection

Seventy seedlings (lines) derived from seeds were analyzed for micropropagation in MS+BAP (at 22.2 µM or 5 mg L^−1^) medium. As a response to BAP, we found that 50% of the lines were individualized. Since this type of material is unsuitable for producing shoots, it was immediately discarded.

As we observed in Figure 2, few clonal lines fulfilled the selection characteristics such as appearance/shoot morphology, tissue coloration, spontaneous rooting, malformations, and susceptibility to oxidation. Among these lines, AM32 tissue adapted quickly and presented well-defined shoots. This result indicates efficient nutrient intake and enhanced tissue proliferation and differentiation in the in vitro system. We also observed the formation of crystalline shoots in some lines, which are not very useful for micropropagation due to their rapid individualization after rooting. A specific case with this appearance was the AM31 line, which at first displayed crystalline tissue, but then adapted and was selected for propagation. Chloric shoots were also observed in at least nine lines that eventually died. Finally, four lines of seedlings derived from L12 seeds were selected: M06, AM20, AM21 and M14d-e. The M14d-e seedling was more robust and larger, presenting a cotyledon leaf with a length and width of 3.20 and 0.50 cm, respectively, and three developed roots with an average length of 4.2 cm (Table 2). Concerning the L18 lot, six clonal lines were selected, including AM31 and AM32, which yielded similar data regarding cotyledon leaf formation, but only line AM32 exhibited increased root length. Despite having one less secondary root, the AM33 line development parameters were similar to M14d-e, AM31 and AM32.

### 4.3. Shoot Propagation Induced by BAP in MS without Vitamins

Several micropropagation protocols have been designed for *Agave* species. In those protocols, Murashige and Skoog medium has been supplemented with combinations of cytokinins and auxins such as 2,4D (2,4-dichlorophenoxyacetic acid), IBA (Indole Butyric Acid), BA (6-benzyl adenine), BAP (6-benzyl aminopurine), NAA (Naphthalene acetic acid), IAA (Indole-3-acetic acid), TDZ (Thidiazuron), Kinetin and Topolin. Among these, 2,4D-BA or 2,4D-IBA are the most commonly utilized [32,33,34,35,48,49,50,51].

In this work, we compared the shoot multiplication of 10 selected lines using the direct organogenesis method, which involves applying only BAP cytokinin as a shoot inducer in a vitamin-free MS medium for eight weeks. We found that BAP at 5 mg L^−1^ or 22.2 µM produced the highest number of shoots (83.4 ± 4.56) in *A. marmorata* clonal line AM32 and the fewest in line AM24 (6.6 ± 2.07), indicating a heterogeneous response between clonal lines to BAP cytokinin (Figure 3). Our findings are consistent with Arzate-Fernández et al. [52], who reported a heterogeneous response in the shoot multiplication induced by 5 mg L^−1^ of BAP in three explants of *A. marmorata* derived from seeds. In that study, the best multiplication was observed with 22.3 shoots using the meristematic zone of seedlings as an explant. Additionally, rooting was observed in three lines (AM21, AM24, and AM34) with a tendency to be easily individualized. However, for plant production purposes, rooting during shoot multiplication attenuates shoot multiplication and/or upregulates cellular processes to sustain plant growth, consequently increasing production costs.

For in vitro propagation of Agave, the BAP cytokinin alone or combined with other substances has produced different efficiencies for shoot induction. For example, Puente-Garza et al. [46] found that *A. salmiana,* through a selection of seedlings 60 d after seed germination formed 3 and 3.5 shoots with BAP concentrations of 0.5 and 10 mg L^−1^, which correspond to the minimum and maximum BAP concentration in MS + vitamins. The shoot induction increases when BAP (10 mg L^−1^) was combined with 0.04 mg L^−1^ of 2,4D-BA, yielding 14 ± 0.70 shoots. A significant number of shoot buds (32.8 ± 4.2) was reported by Rios-Ramirez et al. [34] using stem tissue from *A. angustifolia* and a BAP and IAA combination (4:1 mg L^−1^). Moreover, Dominguez et al. [32] showed the same induction and inhibition behavior of BAP alone using a protocol for mescal and ornamental agave shoot multiplication. The highest shoot number was observed in *A. cupreata* (10.5) at a BAP concentration of 1.5 mg L^−1^. For shoot induction in agave species *A. difformis*, *A. karwinskii*, *A. obscura* and *A. potatorum*, the optimal BAP concentration was 1.5 mg L^−1^, producing shoots ranging from 4.7 (*A. obscura*) to 6.4 (*A. difformis*). Furthermore, 13.32 µM BAP in MS and MS modified with half nutrient formulation induced 18.5 ± 1.9 and 13.4 ± 0.5 shoots in *A. americana* seedlings after 50 days [52].

BAP is an aromatic cytokinin with good shoot induction because it is quickly metabolized in plant tissues. However, it has side effects at high concentrations, such as decreasing the shoot number or length and inducing apoptosis in plant tissues and cell cultures [53,54,55]. In in vitro carrot cultures, concentrations above 20 µM lead to programmed cell death (PCD) and accelerated senescence (mentioned by Kuniskowska et al. [56]). It has also been shown that BAP reduces cell growth and blocks cell division. We observed a heterogeneous response of *A. marmorata* clonal lines to the same BAP concentration. For example, AM24 exhibited attenuated shooting and rapid induction of tissue senescence, while M14d-e displayed reduced shoot numbers without senescence.

Recently, there has been an emerging interest in studying plant responses to cytokinins. These hormones are perceived via a phosphorelay similar to the two-component systems bacteria use to sense and respond to environmental stimuli. In plants, cytokinin responses involve two-component perception systems (e.g., histidine kinases and histidine phosphotransferases) responding to biotic and abiotic stresses [57,58]. Therefore, both in vitro cultures and BAP and other cytokinins could be employed as a model to study senescence and potential responses of micropropagated plants in hostile environments.

Our results reveal differences in shoot induction and multiplication among the clonal lines, even under identical in vitro culture conditions. From the plant production perspective, the three selected clonal lines (AM31. AM32 and AM33) tolerated a nutrient-free environment such as vermiculite and beach sand. Therefore, the behavior of these lines could contribute to *A. marmorata* mass propagation and conservation.

### 4.4. Total Flavonoid, Phenolic and Saponin Content in the 3 Clonal Lines

In order to find some differences between the selected lines of *A. marmorata*, we quantified the content of flavonoids, phenolic compounds and saponins. These phytochemicals are high-value co-products in agaves [59,60,61]. No statistically significant differences were found between lines concerning the phytochemical content and only slight differences in the flavonoid content in the AM31 line. The steroidal saponin content was slightly elevated in AM32 (12.71 ± 3.87 mg Diosgenin/g extract dried). These values are consistent with data from previous studies with other *Agave* species, from plants grown in the field or via in vitro micropropagation [46,59,62]. However, the total saponin content was higher in AM33. Conversely, we found significant differences in the flavonoid content of *A. marmorata* “hijuelo” maintained in pots for three years compared to the clonal lines (Table 2). This observation is interesting because, physiologically, the synthesis of phytochemicals such as phenolics, flavonoids and saponins has been associated with the plant response to biotic and abiotic stresses, such as drought or the attack by microorganisms and insects [63,64,65].

Currently, in vitro propagated plants are considered a tool for secondary metabolite production because they provide rapid and continuous bioactive compound production, which can be manipulated by growth regulators such as the BAP [66,67,68,69,70]. Riaji et al. [71] observed enhanced total phenolic and flavonoid content in micropropagated plants compared to the mother plant. The three phytochemicals measured in this work are associated with antioxidant activity, as shown in various agave leaf extracts after the harvest of the stem or piña [46]. In vitro protocols through axillary bud proliferation will likely facilitate conservation program development and contribute to sustainable bioactive phytochemical production. Further studies must be undertaken to determine the concentrations and types of phytochemical compounds for predicting if propagated plants will adapt well when reintroduced to the forest.

### 4.5. Mass Spectrometry Identification of Phytochemicals Compounds Accumulated in the Clonal Lines

We employed chromatographic methods to detect differences in the concentrations and types of secondary metabolites accumulated in the shoots of the three *A. marmorata* lines. Based on the percentage of the normalized area (NA) at different wavelengths, differences in the compounds in the extracts were identified (Table 3). Using MALDI-TOF mass spectrometry, we could identify at least 18 molecules of the separated fractions with certainty that presented differences in content between clonal lines (Table 3). Flavonoid molecules included Isorhamnethin Rutinoside, Myricetin 3-O-Glucuronide, Myricetin Glucoside, Myricetin 3-O-Rutinoside, Isorhamnetin-Glucuronide, Isorhamnetin 3-Glucoside, Isorhamnetin-3-O-Diglucoside, Rutin, Quercetin-3-O-Diglucoside, Quercetin-3-Glucoside or Isoquercitrin, and Kaempferol Diglucoside.

We also identified phenolic compounds such as caffeic acid, catechin, gallic acid and sinapic acid. Furthermore, three saponins, possibly tigogenin and hecogenin (Figure 4), were tentatively identified because their standards were unavailable. However, the ionization fragments (*m*/*z*) support the presence of these molecules. Those compounds were previously reported by Puente-Garza et al. [64,72].

Finally, anthocyanins, including Cyanidin 3,5 Diglucoside, Delphinidin 3,5-O-Diglucoside and Delphinidin-3-O-Glucoside, stood out as compounds with the most significant differences between lines. These molecules have been previously reported in several species of wild agaves and attributed to plant protective functions. The presence of anthocyanins in *A. marmorata* is a phenotypic characteristic that distinguishes it from other *Agave* species and is present in an early stage of plant age.

Typically, MALDI-TOF mass spectrometry is used to study large biomolecules (e.g., proteins, lipids, nucleic acids, carbohydrates), and ESI-MS is used to analyze the chemical profile of low molecular weight molecules (<1000 Da). However, this paper demonstrates that operating the MALDI-TOF-MS in the reflectron mode facilitates the assessment of flavonoids, phenolic compounds, and anthocyanins in *A. marmorata* shoots. Indeed, previous studies reported the usefulness of MALDI-TOF-MS in profiling phenolic compounds in plants [73,74,75] and detecting anthocyanins (*m*/*z* 250–800 Da) [76]. Ionization fragments with *m*/*z* between 484–656 Da were associated with glycosylated anthocyanins, and those with *m*/*z* 295, 303, 325, 482, 484, 496 and 595 corresponded to glycoside flavonoids with hexoses and pentoses such as Quercetin, Myricetin and Kaempferol [77]. Additionally, ionization fragments with *m*/*z* > 1000 Da corresponded to saponins [72].

### 4.6. Compatibility of Three Clonal Lines with Endophytic Bacteria

Endophyte microorganisms (bacteria and fungi) have been recognized for their role in plant growth and protection against abiotic and biotic stress [78,79]. Plant-microorganism associations date back more than 400 million years. Ecologically, plants are considered holobionts, where endophytic microorganisms are essential to the plant microbiota. Some reports emphasized the role of endophytes in micropropagated plants through “biotization”, in which their inoculation improves seedlings’ growth, development and resistance, even facilitating adaptation to safe transfer under ex vitro conditions at different stages before introduction into the forest [80,81,82]. Prior to this study, our research group demonstrated how endophyte bacteria support *A. tequilana* growth under limited nitrogen conditions [37].

Our findings indicate incompatibility between *E. cloacae* and the clonal lines and specificity of *A. xylosoxidans* and *P. aeruginosa* for *A. marmorata*. According to Sneck et al. [83], one of the potential consequences of vertical transmission is the specialization between the host and its symbiont (endophyte). These symbionts may have become compatible with host species or genotypes and are incompatible with genetically novel hosts. Accordingly, host colonization failures and poor symbiont growth in plants genetically different from the mother plant (e.g., seed-derived plants) will be observed. The general definition of endophytes refers to bacteria and fungi that can live in the plant tissues without symptoms of disease infection. However, the phenotypic plasticity of endophytes explained their conversion from mutualism to antagonism behavior in different plants according to the hypothesis proposed by Schulz and Boyle [84].

The *E. cloacae* is an endophyte widely distributed among plants and their seeds [85,86,87,88]. Results obtained by our research group showed that *E. cloacae* SEA02 induced a noticeable biomass accumulation in micropropagated *A. tequilana* plants, providing organic nitrogen once the bacterium colonized the root (Garcia-Ochoa manuscript in preparation). Additionally, in micropropagated banana plants, highly colonizing endophytic *E. cloacae* (C2 strain) inoculation enhanced plant growth, transferring nutrients from the soil to the roots, thus, making it an essential bacterium for bananas [89].

Previously, it was demonstrated that inoculating the roots of *A. tequilana* with *E. cloacae* SEA02 rapidly induces H_2_O_2_ production in the whole plant in just a few minutes [84]. This fact is linked to a natural response of plants to endophyte perception, where plants consume bacteria to obtain nutrients under limited nutrient conditions, especially nitrogen [37,86].

*B. tequilensis* is an endophytic bacterium contained in the seeds of *A. tequilana*. It greatly affects the AM33 line, especially root formation (number and length). In the other clonal lines, its effects were quite limited. It has been demonstrated that when grown under limited N and P, *B. tequilensis* provides organic N to *A. tequilana* plants [37]. *B. tequilensis* has been shown to transfer organic N by isotopically marked proteins and nucleic acids of bacteria using ^15^NH_4_Cl as the nitrogen source. After several inoculations, the nitrogenous material of *A. tequilana* cells began to incorporate ^15^N, even in the chlorophyll, where the molecule’s four nitrogen atoms are isotopically labeled [37]. Herein, the AM32 line had a moderate response to the inoculation of this bacterium, and some plants died.

The effects of bacterial inoculations on the plant growth of each clonal line are summarized in Figure 6, which also indicates the best growth parameter caused by one specific bacterium. As mentioned previously, those plants inoculated with *E. cloacae* resulted in more dead plants; thus, it is difficult to ascertain whether *E. cloacae* behaves like a pathogen or does not support the growth of plants since no signs of bacterial rot or disease were observed and no significant differences in death among the three clonal lines. According to Garcia-Ochoa from our research group (manuscript in preparation), the *E. cloacae* SEA02 strain has plant-promoting properties such as nitrogen fixation and ACC deaminase activity. However, this endophyte does not produce auxins, likely accounting for the lack of root stimulation in *A. marmorata*, which hindered several growth parameters. In contrast, *Bt*, *Ax* and *Ps* strains produce IAA-type auxins.

We did not evaluate if genetic differences between the selected clonal lines, but we did observe some differences in the plants’ growth responses due to the endophyte inoculation. The highest response was observed on AM32, especially concerning root formation and growth. Root growth induction is essential for agaves because it provides anchorage in the soil and water and nutrient absorption, fundamental processes in eroded and dried soils.

The *A. xylosoxidans* strain contributed the greatest to root formation in all clonally propagated lines, producing 16 new roots. A study by Aguilar-Jimenez and Rodriguez [88] reported root elongation close to 9.3 cm in *A. marmorata* seedlings grown in an MS medium supplemented with 10 mg/L of the Indole Acetic Acid (IAA) auxin. The *A. xylosoxidans* N-strain used in this work is a diazotrophic endophyte, and a phosphate solubilizer previously isolated from *A. marmorata* seedlings developed 10 d after seed germination [89].

Few published studies have demonstrated the potential of *A. xylosoxidans* as an endophyte and plant growth-promoting bacteria (PGPB). Benson et al. [90] reported that *A. xylosoxidans* strain AUM54 combined with indole-3-butyric acid (IBA) promoted shoot growth, root length, the number of roots and stress tolerance compared to control plants under micropropagation conditions of the endangered medicinal plant *Naravelia zeylanica*. The authors also showed that *A. xylosoxidans* plays a role in increasing plant survival during greenhouse acclimatization. Another study demonstrated the biological control effect against the fungus *Magnaporthe grisea* exerted by the same strain of *A. xylosoxidans* when inoculated in rice seeds. In addition to decreasing the colonization of the phytopathogenic fungus, there was an increase in the germination rate, seedling vigor index and yield of inoculated rice plants without disease conditions [91].

Furthermore, Jana and Yaish [92] recently demonstrated that *A. xylosoxidans* SQU-1, rhizobacteria strain isolated from date palms (*Phoenix dactylifera* L.) grown under salinity, enhanced date palm seed germination and *Arabidopsis* growth under normal and saline conditions. The dry weight gain in AM32 line plants is possibly related to the contribution of *Ax* to nutrient supply. However, the mechanisms and responses of the clonal line to biotization with *A. xylosoxidans* still need to be explored.

Plant micropropagation often results in differences in the endophyte microbiome compared to plants grown conventionally. Indeed, many beneficial microorganisms are eliminated through the micropropagation protocols used. Biotization with endophytes is an attractive new strategy to protect plants that are micropropagated, avoiding mortality when transferred from in vitro to ex vitro conditions. Micropropagated plants inoculated with endophyte bacteria exhibit physiological and developmental changes and pathogen resistance [93,94,95,96]. It is evident that reintroducing these bacteria improves stress tolerance and promotes growth and establishment in the soil. These strategies are undoubtedly becoming attractive tools for the ex-vitro establishment of plants in productive systems [80,81].

## 5. Conclusions

A micropropagation protocol was developed using axillary bud proliferation to multiply clonal lines derived from *A. marmorata* seeds. The application of this process could contribute to improving *Agave* reintroduction into the forest for conservation and ecosystem restoration purposes. We selected the clonal lines based on seed germination capacity, seedling stability, and efficient shoot multiplication in the presence of the BAP cytokinin. The variations in multiplication capacity and stable tissue formation observed in different clonal lines may indicate genetic variability in seed populations. These genetic variations could be exploited for the production of high-quality plants from seeds with different capacities, such as fast propagation, higher adaptation to stress, and growth under nutrient limitation, but allows for maintaining different genetics in the plantations to satisfy the demand for plants without putting a natural resource at risk.

We sought to find other differential features between lines through microbiological approaches and phytochemical analysis. For example, the AM32 line presents a differential response to inoculation with seed-vectored endophytic bacteria, and AM31 differentially accumulates phenolics. The responses to inoculation highlighted the compatibility of *A. xylosoxidans*, an endophytic bacteria of *A. marmorata*.

Furthermore, compatibility and incompatibility responses to inoculation in three of the ten clonal lines indicated plant-bacteria specialization, which could have incompatibility consequences even in plants of the same species but with slight genetic differences. We also identified some phytochemicals considered bioactive compounds accumulated in the *A. marmorata* shoot extracts. This observation opens the possibility of using micropropagation of *A. marmorata* as a source of bioactive compounds, consequently promoting the conservation of this endangered *Agave* species. To our knowledge, this is the first report of secondary metabolites extracted from the vegetal material of *A. marmorata*. Further studies need to be undertaken to determine if these metabolites can be used as biomarkers of plant stage and/or the response to stresses caused by cytokinins or environmental factors.

## Figures and Tables

**Figure 1 biology-11-01423-f001:**
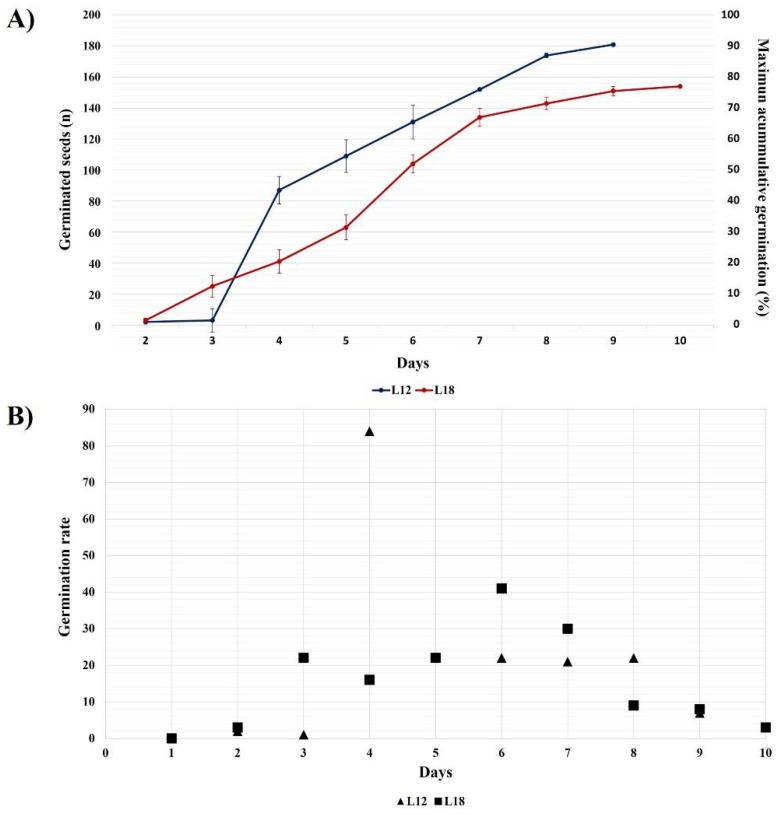
Accumulative germination rate and final germination percentage of the two *A. marmorata* seed lots without plant growth regulators or scarification treatment. (**A**) The number of germinated seeds and the percentage of accumulative germination, and (**B**) the germination rates. The L12 reached the maximum rate on day 4, with 87 seeds germinating, and the final percentage was 90% on day 9. The L18 seeds had the best germination rate on day 6, but the selected seedlings were taken from day 3. The final percentage for L18 seeds was 13% lower than for L12. The data are expressed as the mean ± SD of three independent replicates with 200 seeds per replicate.

**Figure 2 biology-11-01423-f002:**
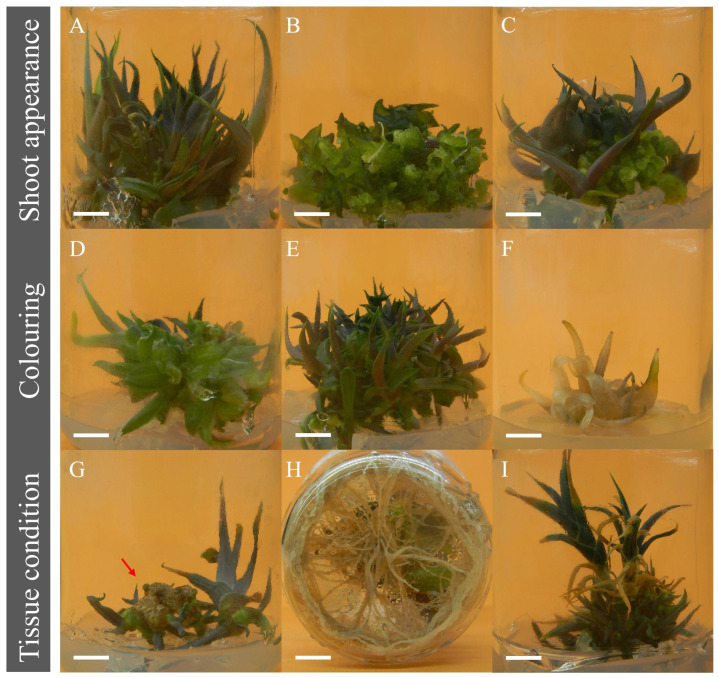
Seedling responses to MS+BAP (at 22.2 µM) for clonal line selection. Shoot appearance: (**A**) Completely formed—AM32; (**B**) Hyperhydric shoots (crystalline)—AM31; (**C**) Completely formed and hyperhydric (combination)—AM31; Coloring: (**D**) Totally green—AM32, (**E**) Purple tones—AM31, AM33, AM34, AM36 and M14D-E; (**F**) White coloring/chlorotic—AM50 to AM58; Tissue condition: (**G**) Crust-browning tissue—AM32 and AM34; (**H**) Rooting—AM20 and AM31; and (**I**) Senescent tissue (shoots/leaves)—AM36, M06 and AM38. Bar = 1 cm.

**Figure 3 biology-11-01423-f003:**
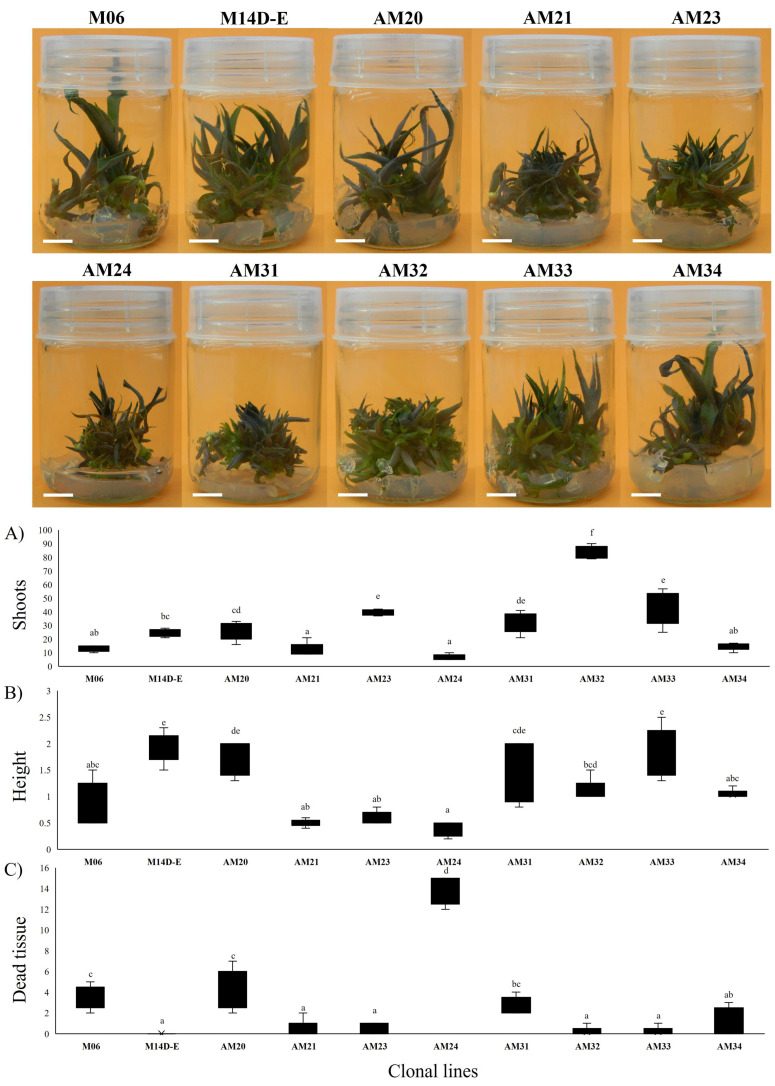
Effect of BAP cytokinin (22.2 µM) on axillary bud proliferation of seed-derived clonal lines of *A. marmorata* on MS medium after 8 weeks of culture. (**A**) Number of shoots formed, (**B**) Shoot height in cm, and (**C**) Number of dry leaves representative of dead tissue. The highest number of shoots was observed in the clonal line AM32, followed by AM33 and AM23. The most stable line was M14d-e which was without senescence tissue. Data are presented as the mean ± SD (top and bottom bars) of results from five repeated experiments. Different letters indicate significant differences, according to Tukey *p* ≤ 0.05. Bar = 1 cm.

**Figure 4 biology-11-01423-f004:**
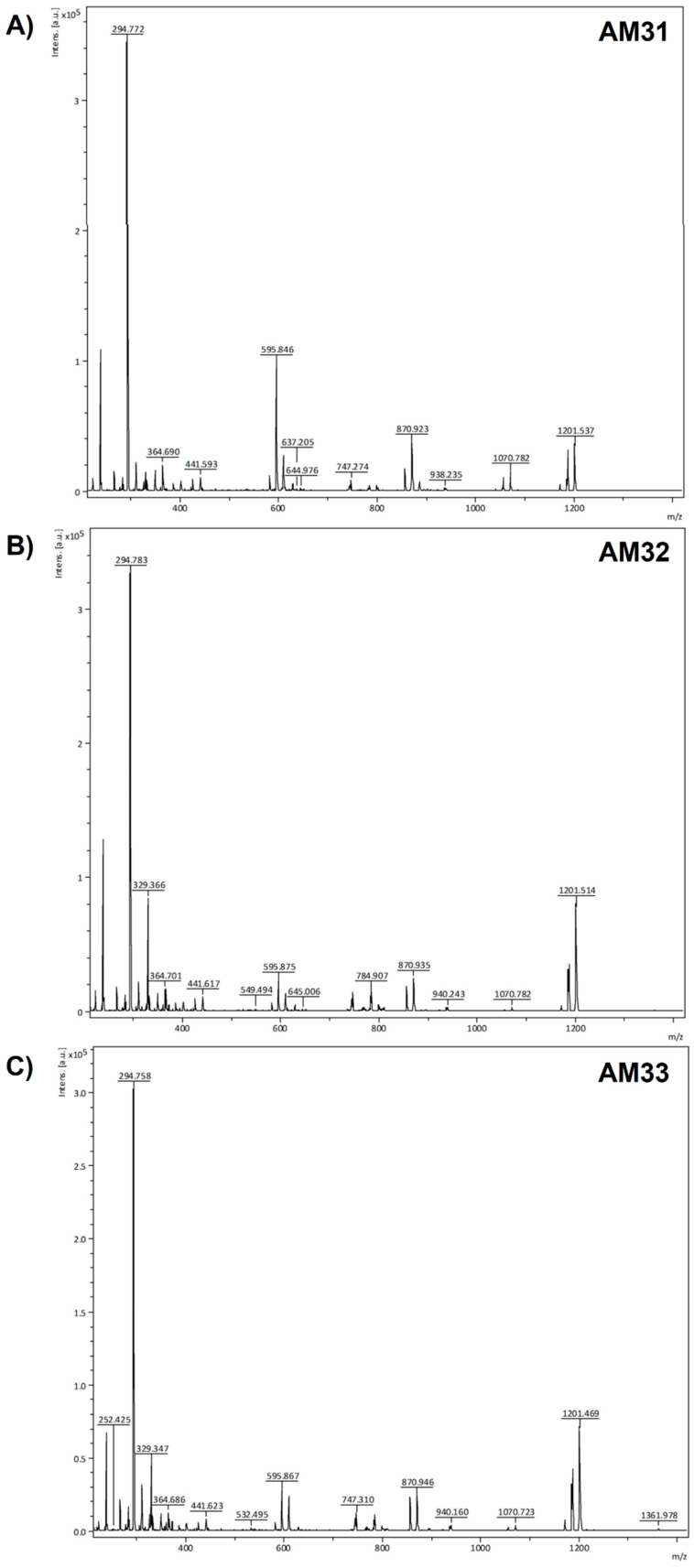
Positive ion MALDI-TOF mass spectra of the hydroalcoholic shoot extracts of the three clonal lines in reflectron mode. (**A**) AM31, (**B**) AM32, and (**C**) AM33. The concentration of each extract was adjusted to 10 mg/mL. The ion fragments at *m*/*z* 254.425 and 294.758 correpond to DHB matrix [M+H]^+^ and [M+K+]^+^.

**Figure 5 biology-11-01423-f005:**
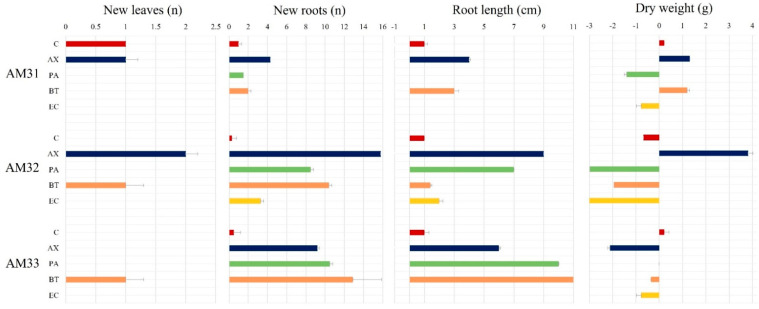
Endophytic bacteria affect the growth of clonal lines AM31, AM32 and AM33 under nutrient limitation. The plant control group was treated with a saline—glucose solution. The positive and negative effects on plant growth must be attributed directly to bacterial inoculation. Data are presented as the mean and standard error of the mean of inoculated plants in forming new leaves, roots, root length and dry weight 30 d post-inoculation. Values and mean represent three replicated experiments. C = control, AX = *A. xylosoxydans*, PA = *P. aeruginosa*, BT = *B. tequilensis* and EC = *E. cloacae*.

**Figure 6 biology-11-01423-f006:**
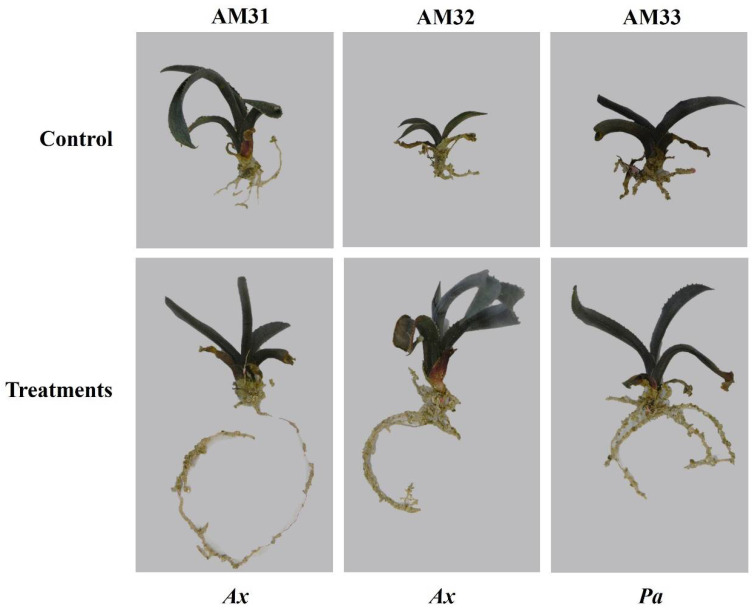
Response of *A. marmorata* clonal lines to endophytic bacteria inoculation. The figure summarizes the best treatment between bacterial inoculation and control, where *A. xylosoxidans* was the best treatment for AM31 and AM32, and *P. aeruginosa* was best for AM33.

**Table 1 biology-11-01423-t001:** Plant growth promoting properties of endopyte bacteria used for biocompatibility assays.

Strains	Endophyte Bacteria	ACC	IAA	N-BRIP	NFB
*Bt*	*Bacillus tequilensis*	+	+	+	+
*Ec*	*Enterobacter cloacae*	+	−	+	+
*Ax*	*Achromobacter xylosoxidans*	+	−	−	+
*Pa*	*Pseudomonas aeruginosa*	+	−	−	+

**Table 2 biology-11-01423-t002:** Parameters of the embryonic leaves, roots and leaves formed in the seedlings from two germination lots selected for multiplication.

				Embryonic Leaf		Seedlings		
Lot	Seed	Germination (d)	Appearance of the Embryonic Leaf (d)	Length (cm)	Width (cm)	Length (cm)	Secondary Roots (n)	Leaves Formed (n)
L12	M06	3	4	3.0	0.3	3.8	2	2
	M14D-E	3	4	3.2	0.5	4.2	3	2
	AM20	3	5	3.0	0.3	3.7	2	2
	AM21	3	5	3.0	0.3	4.0	2	2
L18	AM23	3	5	3.0	0.4	4.0	2	2
	AM24	3	6	3.0	0.3	4.0	2	2
	AM31	3	4	3.2	0.5	4.2	3	2
	AM32	3	4	3.2	0.5	4.3	3	2
	AM33	3	4	3.2	0.5	4.2	2	2
	AM34	3	4	3.0	0.3	4.0	2	2

Description of the nomenclature used. (d): days, (cm): centimeters, (n): number.

**Table 3 biology-11-01423-t003:** Total phenolic, flavonoid and saponin content in crude dried shoot extracts from three *A. marmorata* clonal lines.

Sample	Total Phenolic Content (mgGAE/g Crude Dried Extract) Mean ± SE	Total Flavonoid Content (mg Quercentin E/g Crude Dried Extract) Mean ± SE	Total Steroidal Saponin Content (mg Diosgenin E/g Crude Dried Extract) Mean ± SE	Total Saponin Content (mg Diosgenin E/g Crude Dried Extract) Mean ± SE
AM31	15.28 ± 0.78 ^b^	13.94 ± 0.19 ^b^	10.79 ± 2.84 ^a^	42.46 ± 0.05 ^a^
AM32	12.90 ± 1.38 ^ab^	11.22 ± 1.31 ^b^	12.71 ± 3.87 ^a^	43.59 ± 3.45 ^a^
AM33	13.89 ± 0.35 ^ab^	12.20 ± 2.02 ^b^	10.49 ± 3.81 ^a^	47.66 ± 2.57 ^a^
HAM	11.20 ± 2.36 ^a^	1.67 ± 0.10 ^a^	12.74 ± 0.10 ^a^	48.42 ± 3.78 ^a^

Different letters mean significant differences between extracts, *p* < 0.05.

**Table 4 biology-11-01423-t004:** Retention times (tR), wavelength (λ nm), normalization area percentage (%) and [M+X]^+^ and ion fragments (*m*/*z*) of collected fractions of hydroalcoholic crude extracts of *A. marmorata* shoots using HPLC-UV and MALDI-TOF MS.

Fraction	tR (min)	(λ nm)	Percentage by Normalization Area (%)			*m/z* (+)/Ion Assignment	Tentative Identification	Metabolite Class
			AM31	AM32	AM33			
F1	2.42	205	7.01 ± 4.25 ^a^	24.19 ± 1.57 ^b^	8.15 ± 4.21 ^a^	224, 329, 581 636, 782, 1070, 1200	Unknown	Mixture of compounds
F2	2.95	275	5.17 ± 0.29 ^a^	3.69 ± 0.92 ^a^	3.57 ± 0.69 ^a^	290 [M], 581 [2M+H, dimer], 870 [3M+H, trimer], 1201 [4M+H, tetramer+K]	Catechin	Phenolic
F5	5.37	275	8.28 ± 1.84 ^a^	6.93 ± 0.81 ^a^	6.97 ± 1.67 ^a^	171 [M], 343 [2M+H, dimer], 512 [3M+H, trimer], 684 [4M+H, tetramer]	Gallic Acid	Phenolic
F7	8.47	275	0.73 ± 0.57 ^a^	3.32 ± 0.54 ^b^	3.04 ± 1.27 ^b^	635 [M+H+Rutinoside]	Isorhamnetin Rutinoside	Flavonoid
F8	9.89	275	10.68 ± 6.22 ^a^	1.49 ± 0.32 ^b^	1.90 ± 0.04 ^b^	482 [M+H−Glucuronide]	Myricetin-3-Glucoside	Flavonoid
F12	17.35	275	1.20 ± 0.15 ^a^	5.06 ± 0.56 ^b^	4.69 ± 0.87 ^b^	496 [M+H−Glucuronide]	Myricetin-3-Glucuronide	Flavonoid
F14	19.75	205	2.86 ± 1.19 ^a^	0.77 ± 0.42 ^b^	1.15 ± 0.05 ^b^	628 [M+H+Rhamnosylglucoside]	Myricetin-3-Rutinoside	Flavonoid
F15	20.63	275	0.3 ± 0.17 ^a^	3.46 ± 0.42 ^b^	3.85 ± 1.71 ^b^	180 [M+H], 360 [M+H, dimer], 539 [M+H, trimer],745 [M+H, tetramer]	Caffeic Acid	Phenolic
F18	25.08	518	12.99 ± 2.21 ^a^	15.60 ± 0.51 ^a^	14.37 ± 5.56 ^a^	303 [M], 484 [M+Glycoside+Na]	Delphinidin-3-O-Glucoside	Anthocyanin
F22	27.72	275	1.80 ± 0.70 ^a^	1.06 ± 0.09 ^a^	1.65 ± 0.45 ^a^	607 [M+H]	Rutin	Flavonoid
F23	28.23	518	35.47 ± 2.28 ^a^	23.08 ± 0.25 ^b^	21.62 ± 4.52 ^b^	484 [M+Glycoside+K] 613 [M+2glyc+Na-18]	Cyanidin 3,5-Diglucoside	Anthocyanin
F24	29.45	518	5.12 ± 0.70 ^a^	3.97 ± 1.19 ^a^	4.44 ± 0.42 ^a^	484 [M+Glycoside+Na], 496 [M+Glycoside+K], 656 [M+2Glycoside+K]	Delphinidin-3,5-O-Diglucoside	Anthocyanin
F25	29.87	390	2.08 ± 0.50 ^a^	3.95 ± 1.33 ^a^	4.73 ± 1.23 ^a^	296 [M], 484 [M+H+Glucoronide+Na]	Quercetin-Glucuronide	Flavonoid
F28	34.15	275	2.17 ± 0.33 ^a^	2.88 ± 1.09 ^a^	1.75 ± 0.23 ^a^	224 [M] 455 [2M+H, dimer] 736 [3M+H, trimer]	Sinapic Acid	Phenolic
F29	34.60	275	23.79 ± 0.11 ^a^	12.71 ± 2.98 ^b^	12.26 ± 0.83 ^b^	621 [M+H+2Glucoside]	Quercetin-3-O-Diglucoside	Flavonoid
F30	35.87	275	5.20 ± 0.62 ^a^	3.78 ± 1.53 ^a^	3.71 ± 0.57 ^a^	465 [M+H+Glucoside]	Quercetin-3-Glucoside	Flavonoid
F31	38.17	275	6.53 ± 3.69 ^a^	5.62 ± 0.46 ^a^	5.67 ± 0.80 ^a^	605 [M+H+2Glucoside] 628 [M+H+2 Glucoside+Na]	Kaempferol-Diglucoside	Flavonoid
F32	38.97	275	2.53 ± 0.28 ^a^	0.70 ± 0.11 ^b^	1.15 ± 0.64 ^b^	635 [M+H+2Glucoside]	Isorhamnetin-3-O-Di-glucoside	Flavonoid
F37	49.89	275	3.51 ± 0.33 ^a^	5.05 ± 1.63 ^a^	4.60 ± 0.07 ^a^	1332 [M+H−4hexose-2pentose]	Tigogenin	Saponin
F42	56.38	275	0.55 ± 0.21 ^a^	1.18 ± 0.61 ^a^	1.49 ± 0.86 ^a^	496 [M+H+Glucoronide]	Isorhamnetin-Glucuronide	Flavonoid
F44	59.42	275	0.84 ± 0.71 ^a^	3.63 ± 1.99 ^ab^	4.87 ± 1.55 ^b^	473 [M+H+Glucoside]	Isorhamnetin-3-Glucoside	Flavonoid
F 47	63.43	275	0.32 ± 0.19 ^a^	1.01 ± 0.59 ^a^	1.03 ± 0.51 ^a^	1185 [M+H−3Hexose-2Pentose]	Hecogenin	Saponin
F 48	64.53	275	0.30 ± 0.24 ^a^	0.85 ± 0.47 ^a^	1.14 ± 0.39 ^a^	1056 [M+H−3Hex-Pen]	Hecogenin	Saponin
F62	84.92	205/390	4.06 ± 2.14 ^a^	3.55 ± 0.02 ^a^	2.76 ± 0.51 ^a^	496 [M+H+Glucoronide]	Myricetin-3-Glucoronide	Flavonoid
F66	90.02	390	6.25 ± 2.47 ^a^	9.77 ± 0.17 ^b^	8.01 ± 0.95 ^ab^	287 [M] 595 [M+Rutinose]	Kaempferol Rutinoside	Flavonoid

Means of the percentage by normalization (Area %) followed by the similar letter within the column are not significantly different (*p* > 0.05) according to the Least Significant Difference test.

## Data Availability

All data in this study can be found in the manuscript.

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
