# Peer review of "Micropropagation of Seed-Derived Clonal Lines of the Endangered Agave marmorata Roezl and Their Compatibility with Endophytes"

_biology, 2022, doi:10.3390/biology11101423_

Round 1
Reviewer 1 Report
Detailed comments: Micropropagation of seed-derived clonal lines of the endangered Agave marmorata Roezl and growth biostimulation by their endophyte Achromobacter xylosoxidans
General comments
· The writing style is too verbose and the entire manuscript can be significantly shortened
· The statistical analyses are missing from all tables – this needs to be added throughout (with the differences/similarities presented as superscripts)
· The title requires revision - the study does not provide a micropropagation protocol in the classic sense and the conclusion of biostimulation by the endophyte is flawed
Specific comments
· L22-24: these two sentences are contradictory. If micropropagation reduces genetic variability, then why is this study being done? The justification for undertaking the research is not clearly articulated, one can presume the reasons for the study but the authors need to state this more explicitly
· L35 and throughout: the correct term is ‘multiply’ not ‘multiplicate’
· The introduction fails to adequately contextualise the knowledge gap that the study will address – as highlighted in the discussion, numerous efforts have been made to micropropagate this species so what is the specific goal of the current study?
· L179: provide justification for selection of only 5 mg/l BAP (e.g. citations of publications)
· The above comment raises a technical flaw in the experimental design – there is no control at the multiplication stage, i.e. no treatment with zero BAP – this means that the conclusions cannot be attributed solely to the presence of BAP. The same comment applies to the rooting treatment. Unless the authors can provide this data, this renders the experimental design flawed and publication cannot be recommended. Indeed, a control treatment was evident in the experiment with endophytic bacteria, so it is confusing as to why controls were not applied to all trials.
· L204: The heading for section 2.5 is missing something and does not make sense
· L368: For section 3.1 and throughout the manuscript, it is strongly suggested that instead of using absolute values, percentages should be used for germination and other measurements. This is convention and it is also a more meaningful comparator than actual counted numbers
· L373-375: this sentence is unclear
· L379: provide a justification for using day 4 as the cut off point for selection of lines following germination.
· L386: should be ‘cotyledonary’ not ‘cotyledonal’
· L387: explain what is meant by ‘stagnant’
· L364-386: these three paragraphs could be more effectively summarised – what is the hypothesis and what were the outcomes.
· L398-401: the legend for this figure needs to be corrected for L12 since it is not accurate
· Fig 2: not all captions have magnification bars
· L437: poor sentence construction
· L440 and throughout: explain what is mean by oxidised since this is not clear from the figure or the description in the text
· L446: Fig 2I is not representative of the description in the text since only some leaves are senescent in the image
· Table 1: what is the unit of measurement for ‘appearance of embryonic leaf’ and why are there two columns for length of the embryonic leaf?
· L464: this statement is not accurate since BAP was used for multiplication. No PGRs were used for germination
· L474 and elsewhere: define what is meant by homogenous and heterogenous
· L482: poor sentence construction
· L483-484 and throughout results: were these differences significant or not? Here and throughout, the statistical analysis needs to be considered when results are described or else what is the point of analysing data?
· L498: But M14D-E had almost no dead leaves?
· Fig 3: magnification bars are missing
· L512: Was there a control (zero kinetin) for the experiment with kinetin? Also provide a justification for using this medium formulation here
· L521-524: these two sentences are contradictory
· Fig 4: suggest deleting this figure since the important results are presented in the table
· Fig 5: is this figure necessary?
· Fig 6: there are significant concerns regarding the interpretation of the outcomes from the endophytic bacteria trial – A. xylosoxydans elicits an inconsistent response across lines so the conclusion that it acts as a biostimulant as mentioned in the title is not accurate or reflected in the results.
Author Response
We want to thank Reviewer #1 and Reviewer #2 for their helpful and insightful comments. In the text below, we have provided point-by-point responses/actions (blue text) to each of the reviewers' critiques and concerns (red text).
General comments
Comment:
The writing style is too verbose and the entire manuscript can be significantly shortened
Response/Action:
The introduction was reduced and focused on the main theme in the revised version of the manuscript.
Comment:
The statistical analyses are missing from all tables – this needs to be added throughout (with the differences/similarities presented as superscripts)
Response/Action:
As requested, the statistical analyses and presentation of differences/similarities have been added to all tables.
Comment:
The title requires revision - the study does not provide a micropropagation protocol in the classic sense and the conclusion of biostimulation by the endophyte is flawed
Response/Action:
We modified the title by changing the term "growth promotion" for compatibility induced by inoculation.
It is also worth mentioning that although A. xylosooxidans (Ax) has an overall effect on the lines, at least at this stage, we cannot detect differences between treatments with bacteria, as you observed. For example, larger plants of the AM32 line treated with bacteria for a longer time (6 months) allow us to differentiate between Ax and Pseudomonas treatments. Therefore, we think it is better to emphasize plant-endophyte compatibility.
Specific comments
Comment:
L22-24: these two sentences are contradictory. If micropropagation reduces genetic variability, then why is this study being done? The justification for undertaking the research is not clearly articulated, one can presume the reasons for the study but the authors need to state this more explicitly
Response/Action:
Indeed, the sentence is contradictory. We have modified this in the revised version of the manuscript. In this work, we wanted to highlight the use of seeds as a suitable material for the A. marmorata to maintain genetic variability for conserving this endangered species. Typically, the agave protocols used explant from leaf and meristems of mother plants, and their propagation produces identical clones that diminish genetic variability
Comment:
L35 and throughout: the correct term is 'multiply' not 'multiplicate'
Response/Action:
This typo has been corrected in the revised version of the manuscript.
Comment:
The introduction fails to adequately contextualise the knowledge gap that the study will address – as highlighted in the discussion, numerous efforts have been made to micropropagate this species so what is the specific goal of the current study?
Response/Action:
The introduction has been trimmed and adjusted to the context of the work as suggested. We have described the objective in the text as follows: This work aimed to efficiently obtain and propagate clonal lines of A. marmorata from seeds based on the natural capacities of seed germination to plant establishment (individualization) without physical or chemical force. The total content of flavonoids, phenolics, and saponins was measured in shoots of the three selected lines, and some bioactive compounds were separated and subsequently identified by HPLC-MALDI-TOF. Additionally, we evaluated the biotization of clonal lines with four endophytic bacteria in order to determine plant-microbe compatibility and their effect on plant development.
Comment:
L179: provide justification for selection of only 5 mg/l BAP (e.g. citations of publications)
Response/Action:
The study of Portillo et al., 2007 has been added. We decided to use 5 mg/L because, at the time of the protocol design, it was an average concentration compared to other reports. It also induces the growth of a good number of shoots. Our idea for using BAP at the same concentration allowed us to know the different responses of the lines to distinguish those plants with abilities to proliferate stimulated by the growth promoter rather than determining the best concentration of BAP for shoot induction.
Comment:
The above comment raises a technical flaw in the experimental design – there is no control at the multiplication stage, i.e. no treatment with zero BAP – this means that the conclusions cannot be attributed solely to the presence of BAP. The same comment applies to the rooting treatment. Unless the authors can provide this data, this renders the experimental design flawed and publication cannot be recommended. Indeed, a control treatment was evident in the experiment with endophytic bacteria, so it is confusing as to why controls were not applied to all trials.
Response/Action:
In supplementary Figure 1, we show the plant tissue of AM31, AM32, and AM33 without BAP. The shoot formation was absent, and there was low vigor in their propagules and senescence characterized by tissue oxidation (brown pigment derived from phenolic compounds). The ten lines tested have the same result. Only three lines that had the highest number of shoots were shown in the figure.
Comment:
L204: The heading for section 2.5 is missing something and does not make sense.
Response/Action:
Thank you for pointing this out. The heading has been modified in the revised version of the manuscript. Indeed, in the experimental section 2.5, single shoots were induced by kinetin for rooting. Then, the plants were adapted to a microcosm with sand-vermiculite as the substrate that was subsequently irrigated with mineral MS without sucrose and nitrogen for bacterial biotization.
Comment:
L368: For section 3.1 and throughout the manuscript, it is strongly suggested that instead of using absolute values, percentages should be used for germination and other measurements. This is convention and it is also a more meaningful comparator than actual counted numbers.
Response/Action:
This section has been modified in the revised version of the manuscript.
Comment:
L373-375: this sentence is unclear
Response/Action:
This sentence has been modified in the revised version of the manuscript. It should read "germination rate".
Comment:
L379: provide a justification for using day 4 as the cut off point for selection of lines following germination.
Response/Action:
In this work, we try to recreate seed germination in nature. Therefore, germinated seeds in the first four days with fast sprouting of their radicle are more competitive for water and nutrient uptake. In this sense, the formed seedlings colonized the substrate more efficiently. In our germination system, we used 12 seeds per petri dish; the hydrogen peroxide accumulation can induce the germination of less suitable seeds, forming weak seedlings that do not achieve optimal development. Seeds that do not germinate by water stimulation in an open environmental system are less competent for mature seedling formation. Seedlings derived from germinated seeds on day eight did not survive.
Comment:
L386: should be 'cotyledonary' not 'cotyledonal'
Response/Action:
This type has been corrected in the revised version of the manuscript.
Comment:
L387: explain what is meant by 'stagnant'
Response/Action:
Stagnant indicated that seedlings did not continue to grow without true leaves and root formation.
Comment:
L364-386: these three paragraphs could be more effectively summarised – what is the hypothesis and what were the outcomes.
Response/Action:
The suggested modifications have been included in the revised version of the manuscript.
Comment:
L398-401: the legend for this figure needs to be corrected for L12 since it is not accurate
Response/Action:
The suggested modification has been included in the revised version of the manuscript.
Comment:
Fig 2: not all captions have magnification bars
Response/Action:
Bars were added and correspond to 1 cm.
Comment:
L437: poor sentence construction
Response/Action:
The suggested modification has been included in the revised version of the manuscript.
Comment:
L440 and throughout: explain what is mean by oxidised since this is not clear from the figure or the description in the text
Response/Action:
The term "oxidized" refers to tissue browning, derived from the oxidation and polymerization of phenolic compounds. In some lines' shoots, we observed hard crust formation on the surface with dead cells. Soaking this dead tissue on the surface of the shoot and transferring it to a fresh medium MS+BAP reactivates shooting. Only lines AM31 and AM34 remained active after this "surgery"; therefore, they were included in the pool of lines for multiplication.
Comment:
L446: Fig 2I is not representative of the description in the text since only some leaves are senescent in the image
Response/Action:
A more representative seedling with signs of senescence was added to Figure 2.
Comment:
Table 1: what is the unit of measurement for 'appearance of embryonic leaf' and why are there two columns for length of the embryonic leaf?
Response/Action:
The unit of measure was days (d). We modified Table 1 to remove this error.
Comment:
L464: this statement is not accurate since BAP was used for multiplication. No PGRs were used for germination
Response/Action:
This sentence has been modified in the revised version of the manuscript.
Comment:
L474 and elsewhere: define what is meant by homogenous and heterogenous
Response/Action:
We use the terms heterogeneous as statistically different/homogeneous without statistical differences.
Comment:
L482: poor sentence construction
Response/Action:
This sentence has been modified in the revised version of the manuscript.
Comment:
L483-484 and throughout results: were these differences significant or not? Here and throughout, the statistical analysis needs to be considered when results are described or else what is the point of analysing data?
Response/Action:
Yes, we observed statistical differences between the lines for the different parameters. The analysis has been considered as you rightly proposed.
Comment:
L498: But M14D-E had almost no dead leaves?
Response/Action:
Yes, it is true. M14D-E is an attractive line that we continue proliferating until today because it shows resistance to abiotic and biotic stresses. Moreover, this line survived a fungal attack before shoot multiplication trials. However, it presents a low shoot formation, which makes it a less optimal line for plant production in the biotization trials with the endophytes.
Comment:
Fig 3: magnification bars are missing
Response/Action:
The missing bars were added to Figure 3.
Comment:
L512: Was there a control (zero kinetin) for the experiment with kinetin? Also provide a justification for using this medium formulation here.
Response/Action:
As we have mentioned before, the objective of this study was not to evaluate root production induced by different hormone concentrations. We needed roots to measure the effect of the inoculation of bacteria. We used kinetin at a low concentration for rooting, but it was reinforced with activated charcoal.
Comment:
L521-524: these two sentences are contradictory
Response/Action:
These sentences have been corrected in the revised version of the manuscript.
Comment:
Fig 4: suggest deleting this figure since the important results are presented in the table
Response/Action:
We agree and have removed Figure 4.
Comment:
Fig 5: is this figure necessary?
Response/Action:
Figure 5 shows a difference between AM33 and the other two, a saponin-like compound having an ion fragment at 1361.78 m/z. We are currently identifying this molecule since it appears to have antimicrobial activities.
Comment:
Fig 6: there are significant concerns regarding the interpretation of the outcomes from the endophytic bacteria trial – A. xylosoxydans elicits an inconsistent response across lines so the conclusion that it acts as a biostimulant as mentioned in the title is not accurate or reflected in the results.
Response/Action:
Indeed, A. xylosoxidans only presents an effect on the growth of AM32. For the other lines, this effect was inconsistent. Your comment reoriented our perception; this bacterium is more compatible with the lines, unlike the other endophytes. A. xylosoxidans is found in the seeds of A. marmorata and can be easily isolated in seedlings (Martinez-Rodriguez et al., 2019). We have found that the biotization with this bacterium in ex-vitro AM32 plants influences the survival of plants to cold stress and activates the endophyte microbiome that confers resistance to abiotic stress (unpublished results).

Reviewer 2 Report
Mezcal is a long-traditional Mexican spirit, but recently, it has been gaining immense popularity worldwide. This has impacted negatively on preserving of all Agaves species used for Mezcal production. This manuscript submitted by Martínez-Rodriguez and colleagues addressed the conservation of endangered Agave marmorata, a species collected in nature and used for Mezcal production in Oaxaca, Mexico. Here, the authors report a protocol for in vitro propagation of seed-derived clonal lines of A. marmorata. Although several protocols for massive multiplication for this Agave species have been previously reported, including somatic embryogenesis, Martínez-Rodriguez's strategy is based on using seed-derived clonal lines, which may help keep genetic diversity. Additionally, they analyzed the effects of in vitro culture on secondary metabolite accumulation.
The manuscript is a document well-organized and written; the materials and methods section is adequately described, the results shown are clear and robust, and conclusions are drawn according to the result data obtained. So, in this point I have only some minor points that should be addressed by authors.
Introduction
-Lines 100-101. More examples of the use of in vitro seed-based propagation and genetic diversity should be described. The authors only referred to one case.
Materials and Methods
-Indicate and uniform the information about city, state, and country for all the reagents used, as shown for some equipment (Lines 164, 224). Likewise, for software used (Lines 352 vs 353).
-Line: 233. Use the word extracts instead of samples.
-Line 267,... Check the spelling of the Celsius symbol according to the journal's rules. I think it has to be a space between the number and the symbol. A surfing web cites the following: For degrees of arc, the degree symbol follows the number without any intervening space, e.g., 43°. For temperature, there is a space between the number and the degree symbol, but no space between the degree symbol and the letter that indicates the scale being used, e.g., 100 °C, 212 °F.
Lines 278 vs 279 vs 525. Uppercase for diosgenin? In these lines, the word is written using uppercase and lowercase.
-Lines 328. Split the words: 60C, ACC deaminase.
Lines 235 vs 334. Uniform the spelling of g unit for speed during centrifugation.
Results and discussions
Line 471. Add a comma after shoot length.
Lines 554-565. Uniform the use of uppercase or lowercase for the metabolite names.
Line 582, 823. hijuelo or offset?
Line 651. Extraordinary? Maybe outstanding is more common to describe performance.
Lines 574, 587, 720, and 729. Data not shown? I think this kind of situation weakens the work. If the information included in data not shown is not relevant to the manuscript, I suggest eliminating these phrases.
Line 843. Split %NA.
Lines 846-851. Mixed use of uppercase and lowercase for metabolite names.
Line 911. Use the abbreviation declared previously for Bacillus tequilensis.
Author Response
General comments
Introduction
Comment:
Lines 100-101. More examples of the use of in vitro seed-based propagation and genetic diversity should be described. The authors only referred to one case.
Response/Action:
We have included more examples in the revised version of the manuscript.
Materials and Methods
Comment:
Indicate and uniform the information about city, state, and country for all the reagents used, as shown for some equipment (Lines 164, 224). Likewise, for software used (Lines 352 vs 353).
Response/Action:
We have addressed these concerns in the revised version of the manuscript.
Comment:
Line: 233. Use the word extracts instead of samples.
Response/Action:
We have made this modification in the revised version of the manuscript.
Comment:
Line 267,... Check the spelling of the Celsius symbol according to the journal's rules. I think it has to be a space between the number and the symbol. A surfing web cites the following: For degrees of arc, the degree symbol follows the number without any intervening space, e.g., 43°. For temperature, there is a space between the number and the degree symbol, but no space between the degree symbol and the letter that indicates the scale being used, e.g., 100 °C, 212 °F.
Response/Action:
We have made this modification in the revised version of the manuscript.
Comment:
Lines 278 vs 279 vs 525. Uppercase for diosgenin? In these lines, the word is written using uppercase and lowercase.
Response/Action:
We have made this modification in the revised version of the manuscript.
Comment:
Lines 328. Split the words: 60C, ACC deaminase.
Response/Action:
We have made this modification in the revised version of the manuscript.
Comment:
Lines 235 vs 334. Uniform the spelling of g unit for speed during centrifugation.
Response/Action:
We have made this modification in the revised version of the manuscript.
Results and discussions
Comment:
Line 471. Add a comma after shoot length.
Response/Action:
We have made this modification in the revised version of the manuscript.
Comment:
Lines 554-565. Uniform the use of uppercase or lowercase for the metabolite names.
Response/Action:
We have made this modification in the revised version of the manuscript.
Comment:
Line 582, 823. hijuelo or offset?
Response/Action:
hijuelo
Comment:
Line 651. Extraordinary? Maybe outstanding is more common to describe performance.
Response/Action:
We have made this modification in the revised version of the manuscript.
Comment:
Lines 574, 587, 720, and 729. Data not shown? I think this kind of situation weakens the work. If the information included in data not shown is not relevant to the manuscript, I suggest eliminating these phrases.
Response/Action:
We have made these modifications in the revised version of the manuscript.
Comment:
Line 843. Split %NA.
Response/Action:
We have made this modification in the revised version of the manuscript.
Comment:
Lines 846-851. Mixed use of uppercase and lowercase for metabolite names.
Response/Action:
We have made this modification in the revised version of the manuscript.
Comment:
Line 911. Use the abbreviation declared previously for Bacillus tequilensis.
Response/Action:
We have made this modification in the revised version of the manuscript.
